

# A new perspective on the taxonomy and systematics of Arvicolinae (Gray, 1821) and a new time-calibrated phylogeny for the clade

Charles B. Withnell[1,2] and Simon G. Scarpetta[2,3,4]

[1] Department of Medical Education/ Anne Burnett Marion School of Medicine, Texas Christian University, Fort Worth, TX, United States of America

[2] Department of Earth and Planetary Sciences/ Jackson School of Geosciences, The University of Texas at Austin, Austin, TX, United States of America

[3] Museum of Vertebrate Zoology, University of California, Berkeley, Berkeley, CA, United States of America

[4] Department of Environmental Science, University of San Francisco, San Francisco, CA, United States of America

## ABSTRACT

**Background**. Arvicoline rodents are one of the most speciose and rapidly evolving mammalian lineages. Fossil arvicolines are also among the most common vertebrate fossils found in sites of Pliocene and Pleistocene age in Eurasia and North America. However, there is no taxonomically robust, well-supported, time-calibrated phylogeny for the group.

**Methods**. Here we present well-supported hypotheses of arvicoline rodent systematics using maximum likelihood and Bayesian inference of DNA sequences of two mitochondrial genes and three nuclear genes representing 146 (82% coverage) species and 100% of currently recognized arvicoline genera. We elucidate well-supported major clades, reviewed the relationships and taxonomy of many species and genera, and critically compared our resulting molecular phylogenetic hypotheses to previously published hypotheses. We also used five fossil calibrations to generate a time-calibrated phylogeny of Arvicolinae that permitted some reconciliation between paleontological and neontological data.

**Results**. Our results are largely congruent with previous molecular phylogenies, but we increased the support in many regions of the arvicoline tree that were previously poorly-sampled. Our sampling resulted in a better understanding of relationships within Clethrionomyini, the early-diverging position and close relationship of true lemmings (*Lemmus* and *Myopus*) and bog lemmings (*Synaptomys*), and provided support for recent taxonomic changes within Microtini. Our results indicate an origin of ~6.4 Ma for crown arvicoline rodents. These results have major implications (*e.g.*, diversification rates, paleobiogeography) for our confidence in the fossil record of arvicolines and their utility as biochronological tools in Eurasia and North America during the Quaternary.

Corresponding author
Charles B. Withnell,
c.withnell@tcu.edu

# INTRODUCTION

Arvicoline rodents (voles, lemmings, muskrats, and their extinct relatives) are the subject of a complex history of taxonomic and phylogenetic research (*Conroy & Cook, 1999*; *Conroy & Cook, 2000*; *Galewski et al., 2006*; *Buzan et al., 2008*; *Robovský, ŘIčánková & Zrzavý, 2008*; *Fabre et al., 2012*; *Martínková & Moravec, 2012*; *Steppan & Schenk, 2017*; *Upham, Esselstyn & Jetz, 2019*; *Withnell, 2020*; *Abramson et al., 2021*). Previous phylogenetic studies focused on subsets of Arvicolinae (*i.e.,* North American *Microtus*; *Conroy & Cook, 1999*; *Martínková & Moravec, 2012*), used only one or two genes (*Buzan et al., 2008*), included arvicolines in larger studies investigating the phylogeny of all rodents (*Fabre et al., 2012*; *Steppan & Schenk, 2017*), or focused on mitochondrial genomes (*Abramson et al., 2021*). A comprehensive combined-evidence molecular and morphological dataset was generated by *Robovský, ŘIčánková & Zrzavý (2008)*. Although a great deal of molecular data is available for arvicoline rodents, no study has synthesized all available molecular data to examine phylogenetic relationships across the group at both the generic and species levels—a new phylogeny has been warranted. Our objectives with this study are two-fold. (1) We wanted to consolidate a current and succinct understanding of the taxonomy and systematics of all the major clades of arvicolines through an analysis of the molecular data available on GenBank. (2) We wanted to use the fossil record to generate a time-calibrated phylogeny for the clade that can serve as the foundation that will allow for quantitative modeling of paleobiogeography and dispersal patterns of arvicolines across Beringia. To accomplish these objectives, we compiled the most taxonomically complete molecular dataset of global Arvicolinae to date (July, 2023), using published nuclear genes and mitochondrial markers, to provide a well-sampled molecular hypothesis of extant Arvicolinae.

## Arvicoline taxonomy and systematics through time

For the sake of simplicity we have organized our summary of arvicoline taxonomy and systematics according to the *Mammal Diversity Database (2023)* of the American Society of Mammalogists. Different tribal affinities have been designated over the years and where relevant we have indicated this change. Tribal affinities in the headings reflect the current (August 2023) understanding recognized by the *Mammal Diversity Database (2023)*.

## Tribe Arvicolini (*Arvicola*)

The number of recognized species in *Arvicola* has varied from one to nine (*Miller, 1912*; *Ellerman & Morrison-Scott, 1951*; *Wilson, Lacher & Millermeier Jr, 2017*; *Kryštufek & Shenbrot, 2022*). The tribe Arvicolini has historically consisted of the members of the current tribe Microtini + *Arvicola*. In a phylogenetic analysis by *Abramson et al. (2021)* *Arvicola* was placed separate (sister to the tribe Lagurini) from the taxa included in Microtini, and therefore the *Mammal Diversity Database (2023)* lists *Arvicola* as the only member of the tribe Arvicolini with the rest of the previous members (*Alexandromys*, *Chionomys*, *Hyperacrius*, *Lasiopodomys*, *Lemmiscus*, *Microtus*, *Mictomicrotus*, *Neodon*, *Proedromys*, *Stenocranius*, and *Volemys*) being moved to the tribe Microtini.

## Tribe Clethrionomyini (*Alticola*, *Anteliomys*, *Caromys*, *Clethrionomys*, *Craseomys*, and *Eothenomys*)

Tribe Clethrionomyini is one of the largest groups of arvicoline rodents with six genera. Taxa that belong to this tribe have a long and complicated history of taxonomic revision (*e.g.*, *Hinton, 1926*; *Miller, 1896*; *Kretzoi, 1969*; *von Koenigswald, 1980*; *Kohli et al., 2014*; *Kryštufek et al., 2020*; *Kryštufek & Shenbrot, 2022*). However, the membership of this tribe has remained fairly stable, but generic affinity of the constituent species has shifted frequently (*Kryštufek & Shenbrot, 2022*). The priority of *Clethrionomys* as the valid genus name for red-backed voles was recently recognized (*Kryštufek et al., 2020*); therefore, we have abandoned the taxonomy used by *Musser & Carleton (2005)* that used the genus name *Myodes*. Historically, species that are now listed in *Craseomys* and *Caromys* have been placed in *Clethrionomys* (*Kryštufek & Shenbrot, 2022*). *Tang et al. (2018)* found a paraphyletic *Clethrionomys* + *Craseomys* with respect to *Alticola*, which supports the hypothesis from *Kohli et al. (2014)* that frequent episodes of hybridization have complicated our understanding of their evolutionary history. The "mountain voles" (*Alticola*) were in the past classified within *Arvicola* and *Microtus*, but both of those hypotheses were demonstrated to be incorrect (*Kryštufek & Shenbrot, 2022*). Recent phylogenetic work with *Alticola* has demonstrated that *Ascizomys*, a taxonomically disputed group, is sister to *Alticola* and often ranked as a subgenus of *Alticola* (*Kohli et al., 2014*; *Bodrov et al., 2016*; *Kryštufek & Shenbrot, 2022*). The "Chinese voles", *Anteliomys*, consists of seven species that have historically been placed in *Eothenomys* (*Liu et al., 2012*). Recently *Anteliomys* was elevated to generic distinction (*Kryštufek & Shenbrot, 2022*). The "Brownish voles", *Caromys*, have been thought of as a clade in between *Microtus* and *Clethrionomys* (*Kryštufek & Shenbrot, 2022*). Today, *Caromys* is considered a valid genus that is sister to *Eothenomys* + *Anteliomys* (*Liu et al., 2012*; *Tang et al., 2018*; *Kryštufek & Shenbrot, 2022*). Red-backed voles belonging to *Clethrionomys* today represent five species. In phylogenetic studies the position of *Clethrionomys* relative to *Craseomys* and *Alticola* has been demonstrated as problematic, most likely due to the recent age of the radiation and frequent hybridization between its members (*Kohli et al., 2014*; *Tang et al., 2018*). Oriental voles, *Eothenomys*, are native to southeast Asia and phylogenetic works have demonstrated their sister position to *Anteliomys* (*Liu et al., 2017*). The exact number of species of *Eothenomys* has been debated and more work is needed to clarify this (*Kryštufek & Shenbrot, 2022*).

From a morphological perspective, species with rooted molars were lumped into Clethrionomyini (*Gromov & Polyakov, 1992*) or the subtribe Myodina (*Pavlinov & Rossolimo, 1998*). Appendicular myological and osteological data support the monophyly of *Alticola* and its close relationship to *Clethrionomys* and *Eothenomys* (*Stein, 1987*). Dental morphology (*i.e.,* small, rooted teeth) alone may indicate that *Clethrionomys* and *Alticola* are early diverging members of Arvicolinae, if rooted teeth are the ancestral condition. *Suzuki et al. (1999)* and *Musser & Carleton (2005)* argued that members of *Clethrionomys* may have independently evolved rooted molar conditions. More recent systematic work using multilocus datasets provided some support for that hypothesis (*Kohli et al., 2014*).

### Tribe Dicrostonychini (*Dicrostonyx*)

Collared lemmings (*Dicrostonyx*) were historically thought to be close to true lemmings (*Miller, 1896*). Early molecular and morphological analyses indicated that *Dicrostonyx* was part of one of the earliest radiations of arvicolines (*e.g.*, *Carleton, 1981*; *Chaline & Graf, 1988*; *Conroy & Cook, 1999*; *Gromov & Polyakov, 1992*). For decades, the dominant viewpoint was that there was a single circumpolar species of collared lemming, *Dicrostonyx torquatus*, but evidence from morphology, genetics, ecology, and karyology indicates multiple species (*Borowik & Engstrom, 1993*; *Eger, 1995*; *Musser & Carleton, 2005*; *Wilson, Lacher & Millermeier Jr, 2017*; *Kryštufek & Shenbrot, 2022*). *Fedorov, Fredga & Jarrell (1999)* further documented the complex biogreographic history for *Dicrostonyx* and the interesting phylogeography of the distinct populations in the Palearctic and Nearctic.

### Tribe Ellobiusini (*Bramus* and *Ellobius*)

*Ellobius*, the Northern mole voles, and *Bramus*, the Southern mole voles are morphologically specialized arvicolines (*Corbet, 1978*; *Pavlinov, Yakhontov & Agadzhanyan, 1995*; *Tesakov, 2008*; *Tesakov, 2016*; *Kryštufek & Shenbrot, 2022*). These genera are remarkable in that their Pleistocene range included parts of Israel and North Africa, areas that no other arvicoline has ever inhabited (or they did not leave a known fossil record; *Jaeger, 1988*). *Arvicola* is a European, fossil-rich genus that was previously hypothesized to be closely related to *Microtus* and within the tribe Arvicolini (*Chaline & Graf, 1988*; *Mezhzherin, Zykov & Morozov-Leonov, 1993*; *Kryštufek & Shenbrot, 2022*).

### Tribe Lagurini (*Eolagurus* and *Lagurus*)

Lagurines have been classified close to lemmings, within *Myodes* and *Microtus*, as its own genus *Lagurus*, and as a tribe or subtribe of Arvicolini (see (*Kryštufek & Shenbrot, 2022*) for a description). Early researchers postulated that the sagebrush vole, *Lemmiscus curtatus* and *Hyperacrius* were members of Lagurini, but both genera have subsequently been removed based on molecular and morphological data (*Abramson et al., 2021*; *Kryštufek & Shenbrot, 2022*). Today, two genera, *Lagurus* and *Eolagurus* are included in the tribe Lagurini (*Gromov & Polyakov, 1992*; *Wilson, Lacher & Millermeier Jr, 2017*; *Kryštufek & Shenbrot, 2022*).

### Tribe Lemmini (*Lemmus, Myopus*, and *Synaptomys*)

Bog lemmings (*Synaptomys* and/or *Mictomys*) have many morphological and molecular characters thought to associate the group with the 'true lemmings' (*Myopus* and *Lemmus*) (*Abramson, 1993*; *Abramson, Petrova & Dokuchaev, 2022*; *Carleton, 1981*; *Chaline & Graf, 1988*). Bog lemmings are a strictly North American clade in the modern biota, but based on the fossil record were hypothesized to have originated c. 4 Ma in Europe, with subsequent dispersal through Beringia into North America (*Repenning & Grady, 1988*). Some paleontologists posited that the northern bog lemming (*Synaptomys borealis*) should be placed in its own genus, *Mictomys*, based on dental morphology (*Repenning & Grady, 1988*). Neontologists (*Hall, 1981*; *Musser & Carleton, 2005*) argued that, at best, *Mictomys* is a subgenus of *Synaptomys* based on morphology and ecology.

The 'true lemmings' *Myopus* and *Lemmus* are thought to be early diverging arvicolines (*Carleton, 1981*; *Chaline & Graf, 1988*; *Abramson, 1993*). The monophyly of the 'true lemmings' + *Synaptomys* (excluding *Dicrostonyx*) was supported by cladistic analysis of allozyme data (*Mezzherin, Morozov-Leonov & Kuznetsova, 1995*), nuclear DNA (*Modi, 1996*), and mitochondrial DNA (*Conroy & Cook, 1999*). The taxonomic treatment of *Myopus* has been complicated. Originally, *Chaline (1972)* treated *Myopus schisticolor* as a species of *Lemmus*. Subsequently, *Chaline et al. (1989)* and *von Koenigswald & Martin (1984)* cited molar similarity between *Myopus* and *Lemmus* and placed *Myopus* as a subgenus within *Lemmus*. Karyotype, body size, fur coloration, other morphologies (skull, feet, and eyes), habitat, and behavior were later invoked to demonstrate that *Myopus* is readily distinguishable from *Lemmus* (*Jarrell & Fredga, 1993*). Therefore, *Musser & Carleton (2005)*, *Wilson, Lacher & Millermeier Jr (2017)*, *Kryštufek & Shenbrot (2022)*, and the *Mammal Diversity Database (2023)* treated it as a separate genus.

### Tribe Microtini (*Alexandromys, Chionomys, Hyperacrius, Lasiopodomys, Lemmiscus, Microtus, Mictomicrotus, Neodon, Proedromys, Stenocranius*, and *Volemys*)

*Alexandromys* is an historical term used to describe the grass voles (*e.g.*, *A. oeconomus*). For most of their named history its members have been classified into *Microtus* and/or *Neodon* with *Alexandromys* often being used as a subgenus (*Gromov & Polyakov, 1977*; *Musser & Carleton, 2005*). Karyology indicated that it might be more complicated (see *Kryštufek & Shenbrot, 2022* for an overview) and recent phylogenetic work has demonstrated *Alexandromys* to be distinct and a valid genus (*Bannikova et al., 2010*; *Haring et al., 2015*; *Steppan & Schenk, 2017*). *Lissovsky et al. (2018)* provided new insight into the monophyly of *Alexandromys* and their recent and complex radiation. The systematic relationships of the snow voles, *Chionomys*, are historically controversial (*Gromov & Polyakov, 1992*; *Yannic et al., 2012*). Some researchers posited that *Chionomys* is a member of Myodini (*Mezzherin, Morozov-Leonov & Kuznetsova, 1995*), just outside of *Microtus* (*Yannic et al., 2012*), and others argued based on known fossils that *Chionomys* is closely related to *Clethrionomys* (=*Myodes*) (*Kretzoi, 1969*; *Chaline, 1987*). Some researchers place *Chionomys* closer to *Microtus* in the tribe Arvicolini (*Kryštufek & Shenbrot, 2022*) although others elevated the subtribe Microtini to a tribe and classified *Chionomys* there (*Mammal Diversity Database, 2023*). High-altitude inhabitants of the Himalayas, Kashmir voles of the genus *Hyperacrius*, were originally thought to be a subgenus of *Microtus* (*Miller, 1896*). However, *Hinton (1926)* named *Hyperacrius* a valid genus that is closely related to but outside of *Microtus*.

Members of *Lasiopodomys* were considered by paleontologists to be the remnants of a group that was previously more speciose and widespread (*Gromov & Polyakov, 1992*; *Repenning, 1992*). We note that the fossil *Lasiopodomys* referred to by *Repenning (1992)* in North America is not the same as the extant Eurasian taxa, further adding to the taxonomic confusion of the genus (*Repenning & Grady, 1988*). Neontologists and paleontologists have recognized the morphological uniqueness of *Lasiopodomys*, but in one allozyme analysis, *Lasiopodomys brandtii* was grouped with *Microtis fortis*

and *Microtus* ( =*Stenocranius*) *gregalis*, thus questioning the generic affinity of these species (*Mezhzherin, Zykov & Morozov-Leonov, 1993*). However, *Musser & Carleton (2005)* retained *Lasiopodomys* at the generic level, and *Robovský, ŘIčánková & Zrzavý (2008)* recognized *Lasiopodomys* as the sister to *Stenocranius*. *Lemmiscus* is a monotypic genus that was long considered a subgenus of *Lagurus* in order to segregate New World sagebrush voles from Old World steppe voles (*Carroll & Genoways, 1980*). Morphological and molecular data, however, indicate that *Lemmiscus* may be closely related to *Microtus* (*Carleton, 1981*; *Modi, 1987*; *Abramson et al., 2021*).

The taxonomy and systematics of *Microtus* are complicated and historically difficult to disentangle. Little consensus exists in the literature on how to treat generic-level identifications of fossil *Microtus*, partially because many hypotheses of *Microtus* relationships were based on tooth characters that have limited systematic potential and have undergone rapid evolutionary change (*Guthrie & Matthews, 1971*; *von Koenigswald, 1980*). Combined with the broad Holarctic distribution of the group, poor genetic sampling, and hypothesized recent origination and diversification, and the result has been taxonomic and systematic chaos. Currently, 60 species of *Microtus* are recognized within six subgenera; *Blanfordimys*, *Euarvicola*, *Iberomys*, *Microtus*, *Pitymys*, and *Terricola* (*Kryštufek & Shenbrot, 2022*; *Mammal Diversity Database, 2023*). *Microtus* (*Blanfordimys*) is a geographically isolated group of voles found in south-central Asia (*e.g.*, Afghanistan) (*Shenbrot & Krasnov, 2005*). They have retained dental characters that have been interpreted as pleisiomorphic, but they have inflated auditory bullae and a mastoid region that is so enlarged that it almost projects beyond the occipital condyle, both of which have been interpreted as highly apomorphic (*Gromov & Polyakov, 1992*). This led some researchers to place them as a subgenus of *Microtus* (*Gromov & Polyakov, 1992*; *Kryštufek & Shenbrot, 2022*) while others gave them full generic distinction (*Musser & Carleton, 1993*). *Bannikova, Lebedev & Golenishchev (2009)* and *Steppan & Schenk (2017)* recovered *Blanfordimys* as sister to *Euarvicola* and this should be further examined.

*Euarvicola*, or the field voles, are a Palaearctic group that for most of their history have been recognized as a single species (*M. agrestis*). However, there are now three recognized species (*Pardiñas et al., 2017*; *Kryštufek & Shenbrot, 2022*). *Iberomys* consists of a single species, *M. cabrerae*, native to the Iberian peninsula (*Kryštufek & Shenbrot, 2022*). Historically, *M. cabrerae* has been considered a close relative to *M. arvalis*, however recent studies have left its position unresolved (*Fink et al., 2010*), close to *Stenocranius* (*Cuenca Bescós et al., 2014*) or close to the Nearctic *Microtus* (*Robovský, ŘIčánková & Zrzavý, 2008*; *Martínková & Moravec, 2012*). *Microtus* (*Microtus*) consists of three species "groups" (*Steppan & Schenk, 2017*; *Kryštufek & Shenbrot, 2022*). These groups include: 1. *arvalis* "Grey voles" 2. *socialis* "Social voles" and 3. *shelkovnikovi*. Grey voles are a group of morphologically cryptic species that, based on studies of *cytb*, contains seven species and is sister to the social voles (*Tougard et al., 2013*; *Mahmoudi et al., 2017*; *Golenishchev et al., 2019*). The *socialis* "social vole" group contains one to eight species (*Jaarola et al., 2004*; *Steppan & Schenk, 2017*; *Thanou, Paragamian & Lymberakis, 2020*; *Kryštufek & Shenbrot, 2022*). The *shelkovnikovi* group consists of a single species (*M. shelkovnikovi*) that has been recovered as the sister taxon to the *socialis* group (*Martínková & Moravec,*

*2012*; *Steppan & Schenk, 2017*). *Pitymys* are a large subgenus of *Microtus* that consists exclusively of North American taxa (*Musser & Carleton, 2005*). *Terricola*, or the pine voles, have commonly been considered a subgenus under *Microtus* by neontologists or a genus all of its own by paleontologists (*Kryštufek & Shenbrot, 2022*). All members of *Terricola* are united morphologicaly by the confluence of triangles 4–5 on the upper first molar (*Kryštufek & Shenbrot, 2022*). Molecular systematics has demonstrated the complexity in the taxonomic richness for this subgenus with as many as five groups being recognized (see *Kryštufek & Shenbrot, 2022* for a summary).

*Mictomicrotus* is a recently named monotypic genus (*Kryštufek & Shenbrot, 2022*). *Liu et al. (2007)* originally placed *M. liangshanensis* in *Proedromys* (*P. liangshanensis*). However, morphology and phylogenetic studies demonstrated that *liangshanensis* is not the sister taxa to *Proedromys bedfordi* and therefore warrants a genus of its own (*Chen et al., 2012*; *Steppan & Schenk, 2017*). Voles of the genus *Neodon* are found throughout the mountainous regions of southern Asia. They have a long and complicated taxonomic history, but their close relationship to *Microtus* has been established, although systematic relationships of the genus relative to other arvicolines are still debated (*Musser & Carleton, 2005*; *Pradhan et al., 2019*). Recent taxonomic revision has seen the number of species belonging to *Neodon* grow (*Liu et al., 2017*; *Pradhan et al., 2019*).

Using morphological characters, *Proedromys* was hypothesized to be closely related to *Microtus*, but its diagnostic traits (massive cranium with wide, heavy, and grooved upper incisors and odd molars) were also used to support the hypothesis of a close relationship with extinct genera such as *Allophaiomys* (*Gromov & Polyakov, 1992*; *Repenning, 1992*). Molecular phylogenies also suggest that *Proedromys* is outside of and thus separate from *Microtus*, although with low support (*Chen et al., 2012*). Currently, *Proedromys* is considered a monotypic genus in the tribe Microtini and thought to be closely related to but separate from *Microtus* (*Mammal Diversity Database, 2023*). The narrow-headed voles, *Stenocranius*, have been considered as a subgenus of *Microtus* as well as *Lasiopodomys* (*Kryštufek & Shenbrot, 2022*). For many decades only one species (*S. gregalis*) was recognized, but recent molecular work has detected a cryptic species, *S. raddei* (*Petrova et al., 2015*; *Petrova et al., 2016*). *Volemys* consists of two high-altitude alpine species native to western Sichuan, China (*Liu et al., 2017*). Species of *Volemys* were previously placed in *Microtus* or were found to be closely related to *Microtus*, and previously published phylogenetic analyses of molecular data hinted that the distribution of *Volemys* may be relictual due to geographic (and correspondingly, genetic) isolation during the Late Pleistocene (*Lawrence, 1982*; *Zagorodnyuk, 1990*).

## Tribe Ondatrini (*Ondatra* and *Neofiber*)

*Ondatra* and *Neofiber* are monotypic genera that have the largest body sizes of all arvicolines (both extant and extinct). Historically, they were placed together in the tribe Ondatrini (*Chaline & Mein, 1979*; *Repenning, Fejfar & Heinrich, 1990*) or subtribe Ondatrina (*Pavlinov, Yakhontov & Agadzhanyan, 1995*). Based on allozyme analysis, *Mezhzherin, Morozov-Leonov & Kuznetsova (1995)* concluded that Ondatrina was one of the first groups of arvicolines to diverge from the ancestral arvicoline population

during the late Miocene. Dental morphology, however, led some paleontologists to consider *Ondatra* and *Neofiber* as more distantly related. Although the most obvious similarity is that they are both large (*Carleton, 1981*; *von Koenigswald, 1980*; *Martin, 1974*; *Martin, 1996*), *Ondatra* has rooted molars, and *Neofiber* has rootless molars. Molecular phylogenies support a sister taxon relationship between the genera (*Modi, 1996*; *Fabre et al., 2012*).

### Tribe Pliophenacomyini (*Arborimus* and *Phenacomys*)

Similarities in dental morphology led some researchers to classify *Arborimus* as a subgenus of *Phenacomys* (*Repenning & Grady, 1988*), but others treated *Arborimus* as a separate genus (*Musser & Carleton, 1993*). Another study placed them together in the tribe Phenacomyini (*Zagorodnyuk, 1990*). Others placed *Phenacomys* with *Phaiomys* and other extinct genera (*Repenning, Fejfar & Heinrich, 1990*) or with the tribe Myodini (*McKenna & Bell, 1997*). Both *Phenacomys* and *Arborimus* have primitive molars that retain the plesiomorphic condition of retaining roots on molars, and they lack cementum in the reentrant angles on those molars; therefore some paleontologists argued that *Phenacomys* is an early relict lineage (*Repenning, 1987*). Currently there are two species of *Phenacomys* and three species of *Arborimus* recognized in two distinct but closely related genera united within the tribe Pliophenacomyini (*Wilson, Lacher & Millermeier Jr, 2017*).

### Tribe Pliomyini (*Dinaromys*)

The Eurasian genus *Dinaromys* is monotypic in the extant biota. The plesiomorphic characteristics (*e.g.*, rooted dentition) of *Dinaromys* caused it to be placed in many different groups: subfamily Dolomyinae (*Chaline, 1975*), Tribe Ondatrini (*Corbet, 1978*), Tribe Clethrionomyini (*Gromov & Polyakov, 1992*), or Tribe Prometheomyini (*Pavlinov, Yakhontov & Agadzhanyan, 1995*). To further complicate their systematic status, *von Koenigswald (1980)* found that the lone extant species of the genus, *Dinaromys bogdanovi*, has an enamel microstructure that is unlike any other known extant species.

### Tribe Prometheomyini (*Prometheomys*)

The 'long clawed mole vole', *Prometheomys schaposchnikowi*, is a monotypic species with plesiomorphic characters usually classified in its own tribe (*Gromov & Polyakov, 1992*). This led *Repenning, Fejfar & Heinrich (1990)* to align *Prometheomys* with *Ellobius* in Prometheomyinae, whereas other researchers place *Prometheomys* into Prometheomyini (*Pavlinov, Yakhontov & Agadzhanyan, 1995*; *Pavlinov & Rossolimo, 1998*; *Mammal Diversity Database, 2023*). Whole mitochondrial genomes and subsequent research has indicated that *Prometheomys* is likely a basal arvicoline (*Ibiş et al., 2020*; *Kryštufek & Shenbrot, 2022*).

## MATERIALS & METHODS

### Taxon sampling

Complete sampling of Arvicolinae has been historically challenging due to the high species diversity, global distribution of the clade, and the relative rarity of some species in
museum collections. We attempted to sample all genera ($n = 32$) and species ($n = 178$) recognized by the *Mammal Diversity Database (2023)* in July 2023. We used *Musser & Carleton (2005)*, *Wilson, Lacher & Millermeier Jr (2017)*, and *Kryštufek & Shenbrot (2022)* to inform our taxonomic coverage, however we used the *Mammal Diversity Database (2023)*, the most current and widely accepted database of mammalian taxonomy, when calculating our taxonomic coverage. That resulted in three datasets; (1) a dataset of only taxa with mitochondrial data ($n = 146$), (2) a dataset of only taxa with nuclear data ($n = 107$), (3) a concatenated dataset that includes both mitochondrial and nuclear loci of $n = 146$ species of extant arvicolines, and is the most taxonomically complete dataset to date (August, 2023) for Arvicolinae (82% species and 100% generic coverage). Portions of this text were previously published as part of a thesis (https://repositories.lib.utexas.edu/bitstream/handle/2152/82563/WITHNELL-DISSERTATION-2020.pdf?isAllowed=y{&}sequence=1).

## Concatenated dataset

Molecular data were obtained from GenBank (*NCBI Resource Coordinators, 2016*) (GenBank accession numbers are in Appendix S1A and deposited in Dryad). Three rodents outside of crown Arvicolinae were used as outgroups (*Fabre et al., 2012*), including *Cricetus cricetus*, *Mesocricetus auratus*, and *Neotoma fuscipes*. Five loci were chosen that previously were demonstrated to be useful for rodent phylogenetics (*Galewski et al., 2006*; *Robovský, ŘIčánková & Zrzavý, 2008*; *Fabre et al., 2012*; *Martínková & Moravec, 2012*; *D'Elía, Fabre & Lessa, 2019*; *Upham, Esselstyn & Jetz, 2019*; *Abramson et al., 2021*). We used two mitochondrial markers, Cytochrome b (*Cytb*) and Cytochrome c oxidase subunit 1 (*COI*), as well as the three nuclear markers, growth hormone receptor (*Ghr*) exon 10, iron responsive element binding protein/retinol binding protein 3 (*IRBP/RBP3*) exon 1, and the Breast Cancer gene 1 (*BRCA1*) exon 11. Other genes, such as *ACP5*, have been used in some phylogenetic analyses of arvicoline rodents (*Bondareva et al., 2021a*; *Bondareva et al., 2021b*), but we chose not to include them because their coverage across all of the taxa included in these analyses was relatively low. Whenever possible, vouchered specimens were used and the voucher numbers as well as author contributions are noted in Appendix S1A. This allowed us to increase our confidence in the taxonomic identification of sequences before phylogenetic analysis was completed. In total there were 5,857 base pairs, and each gene had the following coverage across the 149 taxa (146 arvicolines + outgroups): *Cytb* (147 species for 99% coverage with $n = 122$, 82% vouchered), *COI* (64 species for 43% coverage with $n = 44$, 30% vouchered), *Ghr* (90 species for 60% coverage with $n = 47$, 32% vouchered), *IRBP/RBP3* (105 species for 70% coverage with $n = 61$, 41% vouchered), and *BRCA1* (69 species for 46% coverage with $n = 48$, 32% vouchered). Across the entire dataset there was 36.5% missing data. *Stenocranius gregalis* was the most complete across the five genes (5722/5857 nucleotides for 2.3% missing) while *Lemmus amurensis* had the most missing data (356/5857 nucleotides for 93.9% missing).

Sequences were aligned using the iterative refinement algorithm L-INS-I of MAFFT (*Katoh & Standley, 2013*). Aligned nexus files were imported into AliView (*Larsson,*

*2014*) and nuclear protein coding loci were checked for stop codons and trimmed where needed to ensure that they were in the proper reading frame for the first and third codon positions. PartitionFinder 2 (*Lanfear et al., 2017*) was used to partition the dataset (by codon position for the nuclear protein-coding genes) using the Akaike Information Criterion (AIC) (*Burnham & Anderson, 2004*). With all analyses we used GTR+ Γ or GTR + Γ + I molecular substitution models as suggested by PartitionFinder 2 (*Lanfear et al., 2017*) . For the specific model used with each partition see the results of the PartionFinder 2 analysis in Appendix S1B.

## Phylogenetic analyses

We conducted Maximum Likelihood (ML) and Bayesian Inference (BI) analyses of the concatenated datasets, including mitochondrial only (*Cytb* and *COI*), nuclear only (*Ghr*, *IRBP/RBP3*, *BRCA1*), and analyses of all five nuclear and mitochondrial markers, for a total of six phylogenetic analyses. We did not estimate gene trees for the nuclear exons or investigate a species tree approach because of uneven taxonomic sampling within each locus relative to the total taxonomic sample. With more comprehensive taxon sampling across loci, a species tree approach would be highly informative and could possibly help resolve some of the more challenging areas of the tree (*e.g.*, within *Microtus*).

Three analyses were conducted using ML: (1) mitochondrial only (*Cytb* and *COI*) (2) nuclear only (*Ghr*, *IRBP/RBP3*, and *BRCA1*) (3) concatenated mitochondrial and nuclear (*Cytb*, *COI*, *Ghr*, *IRBP/RBP3*, and *BRCA1*). Three analyses were conducted using BI: (1) mitochondrial only (*Cytb* and *COI*) (2) nuclear only (*Ghr*, *IRBP/RBP3*, and *BRCA1*) (3) concatenated mitochondrial and nuclear (*Cytb*, *COI*, *Ghr*, *IRBP/RBP3*, and *BRCA1*). The ML trees were estimated using RAxML v8.2.12 (*Stamatakis, 2014*) on the CIPRES cluster (*Miller, Pfeiffer & Schwartz, 2010*). We used GTR+ Γ or GTR + Γ + I molecular substitution models as suggested by PartitionFinder 2 (*Lanfear et al., 2017*). For ML analyses support values were estimated using 1,000 nonparametric bootstrap pseudoreplicates. Bayesian inference of the partitioned and concatenated dataset was conducted using the Markov Chain Monte Carlo (MCMC) method in MrBayes 3.2.7 (*Ronquist et al., 2012*). The analysis ran for $3.0 \times 10^7$ generations sampled every 1,000 generations and for two separate and independent runs. Beagle was used for high-performance phylogenetic statistical inference (*Ayres et al., 2012*). Results were examined in Tracer 1.7 (*Rambaut et al., 2018*) to ensure that the independent runs reached stationarity and that the effective sample size (ESS) values were >200 for all model parameters. Trees were summarized with majority-rule consensus trees and the first 30% of the samples were discarded as burn-in. All input files for the RAxML and MrBayes analyses are deposited on Dryad (https://doi.org/10.5061/dryad.qrfj6q5cg).

## Time-calibrated analyses

We conducted a time-calibrated BI analysis in MrBayes 3.2.7 using our concatenated mitochondrial and nuclear datasets. We used a birth-death model and an independent gamma rate relaxed-clock (igr), where each branch has an independent rate drawn from a gamma distribution that was empirically derived in MrBayes. The MCMC chain was

run for $3.0 \times 10^7$ generations (sampled every 1,000 generations) for two runs each with four chains. A temperature of 0.1 was implemented and the first 30% of the data were discarded as burn-in. Results of the analyses were visualized in Tracer v1.7 (*Rambaut et al., 2018*) to ensure runs had reached stationarity and that the effective sample size (ESS) was >200 for all model parameters.

*Microtus* is one of the most diverse and rapidly evolving mammalian genera (*Triant & De Woody, 2006*). Many phenotypic characters are convergent among distantly related species, and high genetic variation has been attributed to karyotypic differentiation, with diploid chromosomal numbers ranging from 17 to 64 (*Triant & De Woody, 2006*). *Triant & De Woody (2006)* documented that *Microtus sensu stricto* has a time-corrected mitochondrial rate of nucleotide substitution of 0.08 substitutions per site per million years. This is higher than most other mammals (*e.g.*, *Pan*, *Bos*, *Ursus*) and obviously would affect divergence time analyses (*Triant & De Woody, 2006*). We therefore calculated the rate of evolution for the mitochondrial only, nuclear only, and concatenated mitochondrial and nuclear datasets using R code provided in *Gunnell et al. (2018)*. This code uses a user inputed tree-age along with branch lengths to calculate the rate of evolution within the tree. The mitochondrial rate of evolution was 0.09 substitutions per site per million years, the nuclear rate of evolution was 0.01 substitutions per site per million years, and the concatenated dataset was 0.065 subtitutions per site per million years. This code is included in our Dryad submission (Appendix S1C). Since we used the concatenated mitochondrial and nuclear dataset for time-calibration we chose to use the substitution rate of 0.065 substitutions per site per million years as the mean clock rate prior. We used the MrBayes command 'prset clockratepr' with a mean of −2.72 (natural log of 0.065) and a standard deviation of 0.12 as calculated in the code from *Gunnell et al. (2018)*.

## Node calibration selection

We used four internal node calibrations and a root calibration in our divergence time analyses. Calibrated nodes were constrained as monophyletic and these nodes were selected after non-calibrated phylogenies were produced. For all nodes, there were no suitable fossils available to help establish calibration maxima, so we used exponential calibration priors for each node. For each calibration, the fossil age was used as the offset. R scripts for calculating a suitable mean are in Appendix S1C.

## Calibration 1: Cricetidae (outgroup) (tree root)

We chose as outgroups three muroid rodents previously found to be closely related to Arvicolinae (*Fabre et al., 2012*; *D'Elía, Fabre & Lessa, 2019*). These three species belong to the subfamilies Cricetinae (*Cricetus cricetus*, *Mesocricetus auratus*) and Neotominae (*Neotoma fuscipes*). The split between Arvicolinae and Cricetinae is reported to have occurred during the middle Miocene (*Fabre et al., 2012*). The split between (Arvicolinae, Cricetinae) and Neotominae was hypothesized to be during the early-Miocene (*Fabre et al., 2012*; *Steppan & Schenk, 2017*). We chose to calibrate the root of our tree using the oldest purported fossil neotomid, *Lindsaymys* sp. cf. *L. takeuchii* (*Kelly & Whistler, 2014*; *Martin & Zakrzewski, 2019*). This calibration is anchored by a lower first molar

(m1) housed at the Los Angeles County Museum (LACM 157168). LACM 157168 is diagnosed as belonging to *Lindsaymys* based on. (1) a moderately hyposodont molar that is smaller than the more temporally recent specimens (2) an m1 with an anteroconid that is not bifurcated and positioned close to the metaconid (3) a metalophulid that connects to the protolophulid I at the junction with the anterolophid (4) presence of an entoconid spur and (5) a moderately deep valley between the metaconid and the lingual edge of the anteroconid and anterlophid (*Kelly & Whistler, 2014*). This last feature is key to distinguishing it from other contemporary taxa such as *Abelmoschomys*, *Antecalomys*, *Prosigmodon*, *Bensonomys*, *Baiomys*, *Symmetrodontomys*, and *Jacobsomys* (*Kelly & Whistler, 2014*). LACM 157168 was found in locality LACM 5720 which is thought to be Latest Clarendonian to early Hemphillian (C13-Hh1, ∼9.2−8.7 Ma) (*Kelly & Whistler, 2014*). LACM Locality 5720 is found in the El Paso Mountains within the Dove Spring Formation of the western Mojave Desert, California (*Kelly & Whistler, 2014*). The site lies underneath Dove Spring Ash number 16 dated *via* Ar/Ar at 8.5 ± 0.13 Ma and above Dove Spring Ash number 15 dated *via* fission tract at 8.4 ± 1.8 Ma (*Whistler et al., 2009*). Biochronology and paleomagnetics place this site in Chron C4A, with a maximum age of ∼9.2 Ma (*Kelly & Whistler, 2014*). We chose to use 8.8 Ma as a conservative estimate of the minimum age of the site, since it is a middle point between 8.5 Ma and 9.2 Ma (*Whistler et al., 2009*; *Kelly & Whistler, 2014*). The soft maximum for the node was based on the divergence time analysis of *Steppan & Schenk (2017)*. We used an offset exponential distribution with a minimum age of 8.8 Ma and a mean of 10.8 Ma. 10.8 Ma was chosen as the mean because it produced an 95% upper bound of the distribution at 15 Ma.

## Calibration 2: Lemmini node

The earliest North American bog lemmings (*Synaptomys*) are from the Hagerman Fossil Beds National Monument, Idaho (*Mictomys = Synaptomys vetus*; (*Ruez Jr & Gensler, 2008*). The offset for this node is anchored by a right m1 (lower first molar) housed at the Idaho Museum of Natural History (IMNH 67002/39517) that has radiometric age control (Ar-Ar) of a basaltic tephra located 30 m above the site and dated at 3.79 ± 0.03 Ma (*Hart & Brueseke, 1999*). Interpolation of depositional rates indicates that the age of the fossil from IMNH locality 67002 is ∼3.95 Ma (*Hart & Brueseke, 1999*). IMNH 67002/39517 was identified as *Mictomys vetus* by having evergrowing molars with cementum in the reentrant angles. The m1 also has a posterior loop with three triangles, and an anterior loop (*Ruez Jr & Gensler, 2008*). Triangles 1 and 2 are broadly confluent with the anterior loop and triangle three is joined by the anterior loop near the midline. Triangles 1 and 3 are nearly twice the width of triangle 2. Because *Synaptomys* was paraphyletic in some of our uncalibrated analyses, we used this fossil to calibrate the crown lemming node instead. We used an offset exponential distribution with a minimum age of 3.95 Ma and a mean of 4.74 Ma (see Appendix S1C for this calculation).

## Calibration 3: Ondatrini node

The oldest known species of this clade, *Ondatra minor*, is found in the Hagerman Formation in Hagerman, Idaho at ∼3.6 Ma (*Hibbard, 1959*). All of the fossils at Hagerman are

constrained between two lava flows and ash units that have yielded ages of 4.0 Ma to 3.2 Ma using Ar-Ar dating methods (*McDonald, Link & Lee, 1996*). We anchored the *Ondatra + Neofiber* node using a left m1 tooth of *Ondatra minor* (USNM 21830) from Hagerman. This m1 was identified as *Ondatra minor* by its relatively large size as well as being rooted and having a posterior loop, five alternating triangles, with a fifth triangle opening broadly into the anterior loop (*Hibbard, 1959*). We used an offset exponential distribution with a minimum age of 3.2 Ma and a mean of 4.9 Ma. The age of 3.2 Ma was chosen because it is the most conservative estimate of the age of the two ash layers described from Hagerman Idaho and deposition interpolation information were not available for the locality.

## Calibration 4: Pliophenacomyini node

Extant voles of the genera *Phenacomys* and *Arborimus* are today restricted to North America. Eurasian specimens of *Phenacomys* were identified from Krestovka, Kolyma Lowland Russia (*Sher et al., 1979*; *Zazhigin, 1997*) and Romanovo 1c, Western Siberia, Russia (*Smirnov, Bolshakov & Borodin, 1986*; *Borodin, 2012*). Recently, a new species (*Phenacomys europaeus*) was described from Europe in Zuurland, the Netherlands, and dated at ~2.1 Ma based on biochronology (*van Kolfschoten, Tesakov & Bell, 2018*). The oldest known record of *Phenacomys*, *P. gryci*, (type locality in the Gubik Formation) is from the Fish Creek fauna of Alaska. The Fish Creek Fauna is in the Gubik Formation, which is an alternating marine and coastal plain sedimentary unit. The Fish creek Fauna is dated at ~2.4 Ma using amino acid racemization ratios, a reversed polarity zone, and the presence of the ancestral sea otter *Enhydrion* and the arvicoline rodent *Plioctomys mimomiformis* (*Carter et al., 1986*; *Repenning et al., 1987*; *Repenning & Brouwers, 1992*). We calibrated the (*Phenacomys + Arboriumus*) node based on the type specimen of *Phenacomys gryci* (a left m1 housed at the United States National Museum USNM 26495). This fossil was assigned to *Phenacomys gryci* by having a rooted m1 that lacked cementum in the reentrant angles. It also possesses a posterior loop, five asymmetrical alternating triangles with a ''*Mimomys* Kante'' on triangle four, and a complex anterior loop (*Repenning et al., 1987*). This node was calibrated using an offset exponential distribution with a minimum age of 2.4 Ma and a mean of 3.27 Ma (see Appendix S1C for this calculation).

## Calibration 5: Ellobiusini node

The timing of the origination and diversification of *Microtus* and its close relatives has been repeatedly contested among paleontologists (*e.g.*, *Repenning, 1992*; *Martin & Tesakov, 1998*). It was argued that the genus *Allophaiomys* gave rise *via* anagenetic evolution to what is recognized today as *Microtus* (*Martin & Tesakov, 1998*), but that hypothesis is controversial (*e.g.*, *Bell et al., 2004*; *Bell & Bever, 2006*). The oldest *Allophaiomys* with external age control is from Hansen Bluff (Colorado) and dated at 1.9 Ma (*Rogers et al., 1992*). The earliest occurrence of *Microtus,* as defined by *Repenning (1992)*, was long thought to be from the Anza-Borrego Desert of California (*Zakrzewski, 1972*), possibly from 1.4 to 1.6 Ma (lacking firm age control). Unfortunately, the specimens from Anza-Borrego had questionable field data; one specimen was found in a fault block and from a different area in the park than originally reported, and a second specimen could not

definitively be assigned to *Microtus* (*Bell & Bever, 2006*; *Murray, Ruez Jr & Bell, 2011*). The oldest known *Microtus* is, therefore, found in the type Irvington Fauna from California dated to 1.21 Ma based on paleomagnetic data (*Bell & Bever, 2006*).

Fossil evidence from *Ellobius*, an early diverging member of the clade that includes *Microtus*, was used here to calibrate the node. The oldest fossils of *Ellobius* are from the Late Pliocene of Kazakhstan and Tajikistan (*Lytchev & Savinov, 1974*; *Zazhigin, 1988*) and the Northern Caucasus (*Tesakov, 2004*). We chose to use a fossil mandible with m1-m3 (Paleontological Institute, Russian Academy of Sciences M-2049/58-KB) of *Ellobius primigenius* from Central Asia (*Lytchev & Savinov, 1974*). This fossil possesses rooted teeth with relatively high crowns, a posterior loop, five alternating triangles, and an anterior loop consistent with *Ellobius* (*Lytchev & Savinov, 1974*). This mandible is part of the Kiikbai Fauna of Kazakhstan dated using biochronology (the occurrence of *Hypolagus brachygnathus*, *Ochotonoides complicidens*, and *Mimomys pliocaenicus*) to the Pliocene at ∼2.4 Ma in the Matuyama Chron (*Sotnikova, Dodonov & Pen'Kov, 1997*). The Kiikbai Fauna is described from the southern flank of the Ilian depression in the Alatau mountains and placed in the European middle Villafranchian land mammal age (*Sotnikova, Dodonov & Pen'Kov, 1997*). We used an offset exponential distribution with a minimum age of 2.4 Ma and a mean of 3.27 Ma (see Appendix S1C for this calculation).

## RESULTS
### Non-clock analyses
Six phylogenetic analyses were conducted on the concatenated dataset that included either all or a subset of the 147 sequences from *Cytb*; 64 from *COI*; 90 from *Ghr*; 105 from *IRBP/RBP3*; and 69 from *BRCA1*. 110 species (74%) included both mitochondrial and nuclear data. 39 species (26%) had only mitochondrial data. GenBank accession numbers are in Appendix S1A. Results from Maximum Likelihood and Bayesian analyses were similar or identical except where discussed below.

### Maximum likelihood (ML) results
A tribe-level summary of the ML tree including only the mitochondrial loci (*Cytb* and *COI*) with rapid-bootstrapping values from RAxML v7.0.4 (lnL = −55,015.19) is presented in Fig. 1A. Moderate support (71–89 BS) was inferred for 3 (8%) of the nodes (Fig. 1A). High support (>90 BS) was found for 2 (6%) of the nodes (Fig. 1A). A species-level tree is presented in Appendix S1D. Moderate support was inferred for 18 (12%) of the nodes (Appendix S1D). High support was found for 51 (35%) of the nodes (Appendix S1D). A tribe-level summary of the ML tree including only the nuclear loci (*Ghr*, *IRBP/RBP3*, and *BRCA1*) with rapid-bootstrapping values from RAxML v7.0.4 (lnL = −21,229.90) is presented in Fig. 2A. Moderate support was inferred for 1 (4%) of the nodes (Fig. 2A). High support was found for 14 (52%) of the nodes (Fig. 2A). A species-level tree is presented in Appendix S1E. Moderate support was inferred for 10 (9%) of the nodes (Appendix S1E). High support was found for 56 (53%) of the nodes (Appendix S1E). Finally, another tribe-level summary of the ML tree including all five loci (*Cytb*, *COI*, *Ghr*, *IRBP/RBP3*, and *BRCA1*) with rapid-bootstrapping values from RAxML

v7.0.4 (lnL = −77,750.23) is presented in Fig. 3A. Moderate support was inferred for 2 (7%) of the nodes (Fig. 3A). High support was found for 9 (32%) of the nodes (Fig. 3A). A species-level tree is presented in Appendix S1F. Moderate support was inferred for 13 (9%) of the nodes (Appendix S1F). High support was found for 72 (49%) of the nodes (Appendix S1F).

## Bayesian inference (BI) results

A tribe-level summary of the BI tree including only the mitochondrial loci (Cytb and COI) with posterior probability values is presented in Fig. 1B. High support (>95 PP) was found for 11 (32%) of the nodes (Fig. 1B). A species-level tree is presented in Appendix S1G. High support was found for 80 (54%) of the nodes (Appendix S1G). A tribe-level summary of the BI tree including only the nuclear loci (Ghr, IRBP/RBP3, and BRCA1) with posterior probability values is presented in Fig. 2B. High support was found for 20 (74%) of the nodes (Fig. 2B). A species-level tree is presented in Appendix S1H. High support was found for 68 (63%) of the nodes (Appendix S1H). Finally a tribe-level summary of the BI tree including all five loci (Cytb, COI, Ghr, IRBP/RBP3, and BRCA1) with posterior probability values is presented in Fig. 3B. High support was found for 17 (57%) of the nodes (Fig. 3B). A species-level tree is presented in Appendix S1I. High support was found for 88 (60%) of the nodes (Appendix S1I).

## Major clade systematic results

The tribe Arvicolini was variably placed in the different analyses (Figs. 1–3). In the mitochondrial analyses (Fig. 1) Arvicolini is weakly placed near the root of the arvicoline tree. In the nuclear and combined analyses (Figs. 2–3) it is placed with stronger support in the derived clade that includes Ellobiusini, Pliomyini, Lagurini, and Microtini. Clethrionomyini is consistently found in all analyses to be a large clade in the middle of the arvicoline tree (Figs. 1–3). Within this tribe there are two clades that are found in all analyses. (1) A clade that consists of *Caryomys*, *Eothenomys*, and *Anteliomys* (Figs. 1–3), and (2) A clade that consists of *Craseomys*, *Clethrionomys*, and *Alticola* (Figs. 1–3). *Anteliomys* is paraphyletic with strong support with respect to *Eothenomys* in the nuclear analysis (Fig. 2) but monophyletic and the sister to *Eothenomys* in the other analyses (Figs. 1 and 3). The paraphyly/monophyly of *Clethrionomys* and *Alticola* vary among analyses (Figs. 1–3). *Clethrionomys* is paraphyletic in the ML and BI mitochondrial and nuclear analyses as well as the combined BI analysis (Figs. 1–3). In the combined ML analysis, *Clethrionomys* was monophyletic (Fig. 3A). *Alticola* was found to be paraphyletic in the mitochondrial analyses (Fig. 1) and the combined ML analysis (Fig. 3A) and monophyletic in the nuclear analysis (Fig. 2) and combined BI (Fig. 3B).

Dicrostonychini is weakly inferred as the sister to Ondatrini in the mitochondrial analyses (Fig. 1), and strongly supported as the sister to Pliophenacomyini in the nuclear and combined analyses (Figs. 2–3). Ellobiusini is consistently found at the base of the large clade that includes Lagurini, Arvicolini (in the nuclear and combined analyses), Pliomyini (in the nuclear and combined analyses), and Microtini (Figs. 1–3). Lagurini is found to be the sister to Microtini in the mitochondrial analyses (Fig. 1) but sister to the

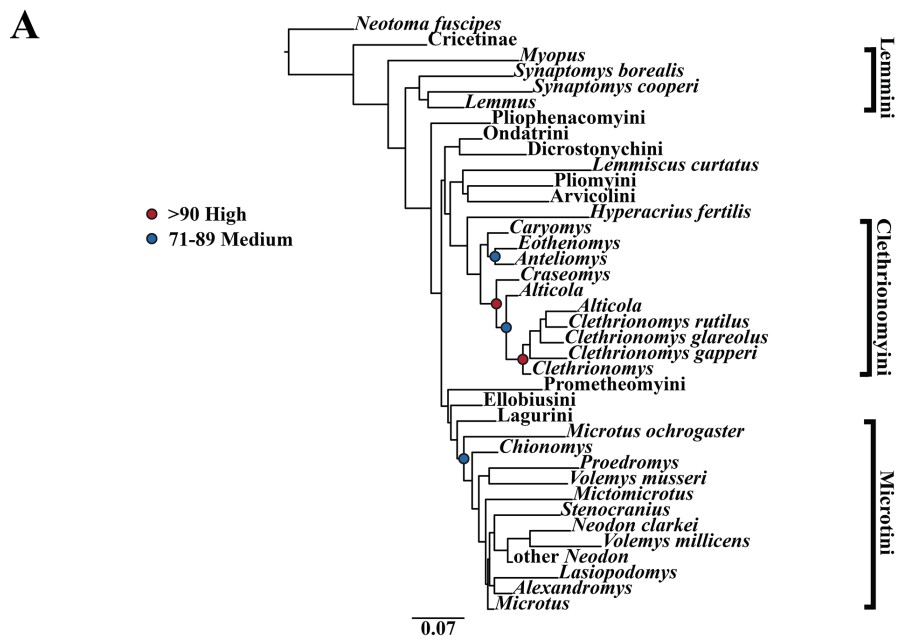

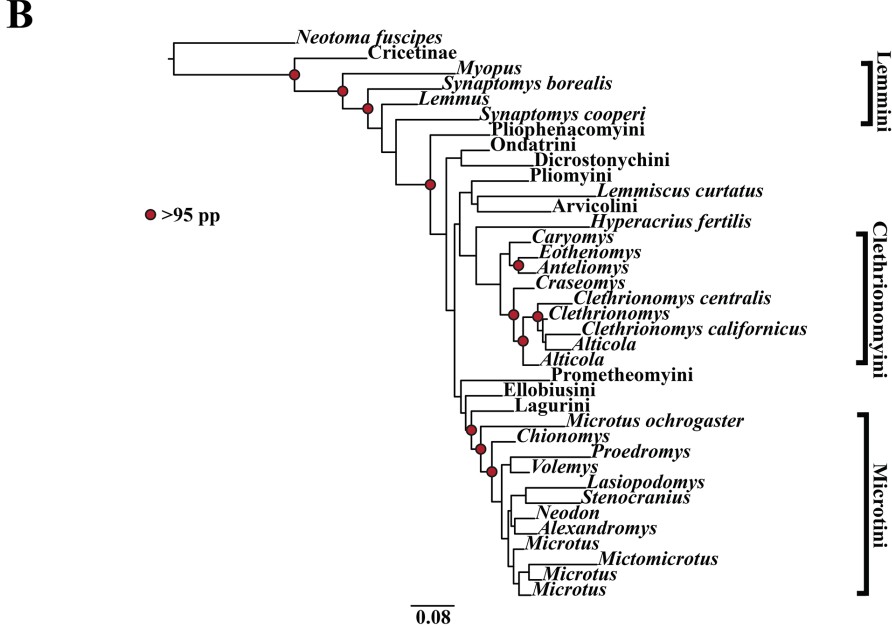

**Figure 1** **Higher-order mitochondrial (*Cytb* and *COI*) only analyses.** (A) Maximum likelihood (ML) tree of mitochondrial dataset comprising 146 arvicolines and three members of Cricetidae. Tree was rooted with *Neotoma fuscipes*. For a species level tree see Appendix S1D. (B) Majority-rule consensus tree produced using Bayesian Inference (BI) methods of the mitochondrial only dataset comprising 146 arvicolines and three members of Cricetidae. For a species level tree see Appendix S1G.

**A**

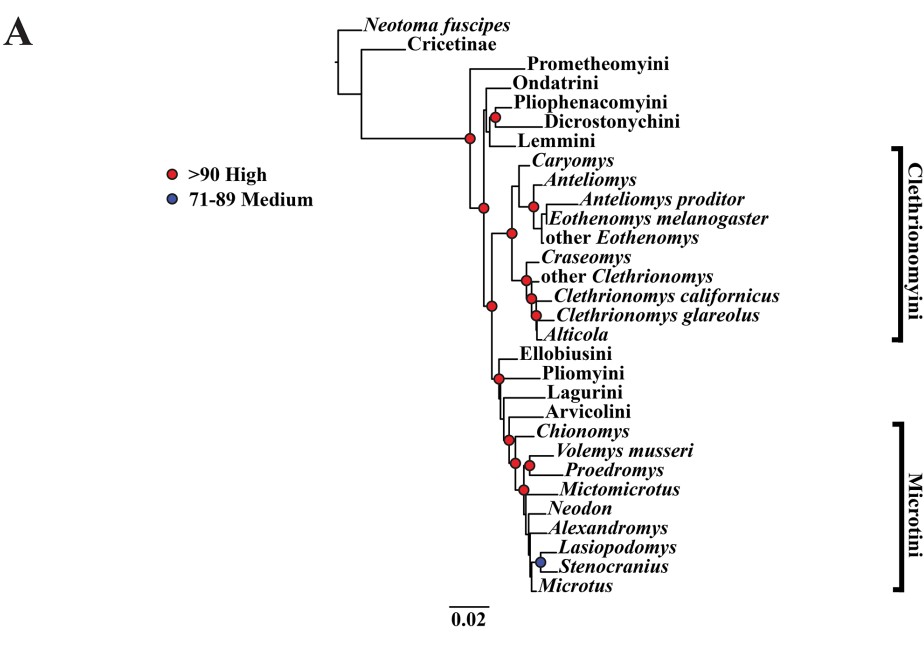

**B**

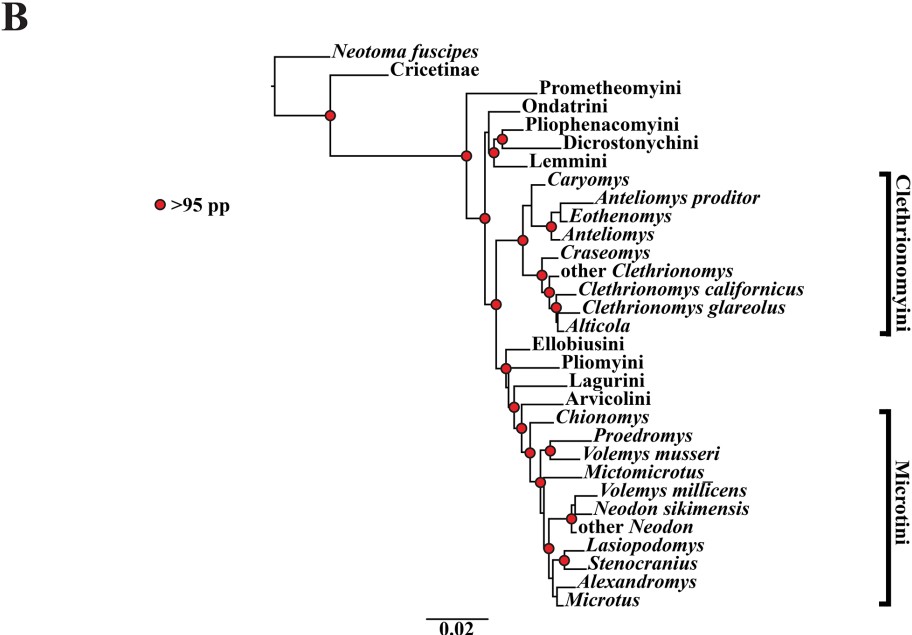

**Figure 2  Higher-order nuclear (*Ghr, IRBP/RBP3,* and *BRCA1*) only analyses.** (A) Maximum likelihood (ML) tree of nuclear dataset comprising 107 arvicolines and three members of Cricetidae. Tree was rooted with *Neotoma fuscipes*. For a species level tree see Appendix S1E. (B) Majority-rule consensus tree produced using Bayesian Inference (BI) methods of the nuclear dataset comprising 107 arvicolines and 3 members of Cricetidae. For a species level tree see Appendix S1H.

**A**

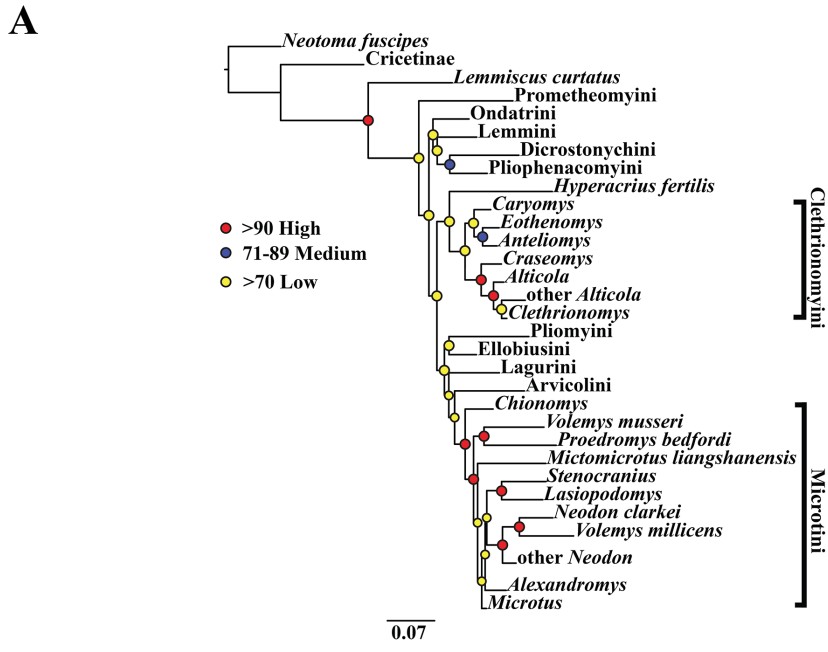

**B**

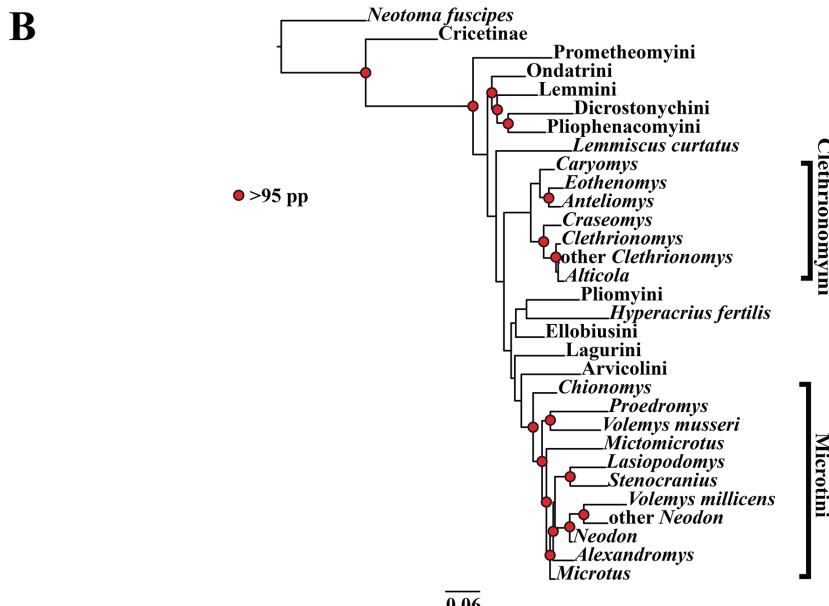

**Figure 3** **Higher-order concatenated mitochondrial and nuclear (*Cytb*, *COI*, *Ghr*, *IRBP/RBP3*, and *BRCA1*) analyses.** (A) Maximum likelihood (ML) tree of concatenated dataset comprising 146 arvicolines and three members of Cricetidae. Tree was rooted with *Neotoma fuscipes*. For a species level tree see Appendix S1F. (B) Majority-rule consensus tree produced using Bayesian inference (BI) methods of the concatenated dataset comprising 146 arvicolines and three members of Cricetidae. For a species level tree see Appendix S1I.

clade containing Arvicolini + Microtini in the nuclear and combined analyses (Figs. 2–3). Lemmini is consistently found near the base of the arvicoline tree (Figs. 1–3), but relationships within the tribe vary across analyses. *Myopus* is the first diverging member of Lemmini in the mitochondrial analysis (Fig. 1 and Appendices S1D and S1G), but earliest diverging 'true-lemming' in the nuclear and combined analyses (Figs. 2–3 and Appendices S1E, S1H and S1F, S1I). *Synaptomys* is paraphyletic or polyphyletic in the mitochondrial analyses (Fig. 1) and combined ML analysis (Fig. 3A) but monophyletic in the nuclear analyses (Fig. 2) and combined BI analysis (Fig. 3B).

Microtini was consistently the most crownward tribe of arvicolines (Figs. 1–3). *Chionomys* is sister to the rest of Microtini in the nuclear and combined analyses (Figs. 2–3), but in the mitochondrial analyses (Fig. 1), a split between *Microtus ochrogaster* and the rest of Microtini was the first divergence. *Volemys* are early diverging within Microtini, and are polyphyletic in the ML mitochondrial analysis (Fig. 1A), nuclear analyses (Fig. 2), and combined analyses (Fig. 3). However, *Volemys* was found to be monophyletic in BI mitochondrial analysis (Fig. 1B). In all analyses, *Proedromys* was placed as sister to *Volemys musseri* (Figs. 1–3). *Lasiopodomys* is the sister to *Stenocranius* in the BI mitochondrial only (Fig. 1B) and the nuclear and combined analyses (Figs. 2–3). In the ML mitochondrial only analysis, *Lasiopodomys* is sister to *Alexandromys*, albeit weakly (Fig. 1A). In the ML mitochondrial only analysis, *Stenocranius* was sister to a paraphyletic *Neodon* (containg a nested *Volemys millicens*) (Fig. 1A), but in the others it is sister to *Lasiopodomys* (Fig. 1B–3). *Neodon* is found to be monophyletic in the mitochondrial BI analysis (Fig. 1B), nuclear ML (Fig. 2A) and paraphyletic in the other analyses (Figs. 1A, 2B, and 3). *Mictomicrotus* is deeply nested within Microtini (Figs. 1A, 2 and 3) or is nested within *Microtus* (Fig. 1B). *Alexandromys* is one of the most nested members of Microtini (Figs. 1–3). *Microtus* is the most nested arvicoline and was monophyletic in the nuclear and combined analyses (Figs. 2–3) but paraphyletic or polyphyletic in the mitochondrial analyses (Fig. 1). Endemic species of North American *Microtus* are monophyletic in the nuclear and combined analyses (Figs. 2–3).

Ondatrini is a rootward tribe of arvicolines in all analyses, but its sister relationships vary (Figs. 1–3). In the mitochondrial analyses Ondatrini is sister to Dicrostonychini (Fig. 1), and in the nuclear and combined analyses it is sister to Pliophenacomyini + Dicrostonychini + Lemmini (Figs. 2–3). Pliophenacomyini is recovered as an early diverging arvicoline sister to Dicrostonychini in the nuclear and combined analyses (Figs. 2–3). Pliomyini is found towards the base of the arvicoline tree in the mitochondrial analyses (Fig. 1), but within the clade that includes Ellobiusini, Lagurini, Arvicolini, and Microtini in the nuclear and combined analyses (Figs. 2–3). Prometheomyini is found at the base of the arvicoline tree in the nuclear and combined analysis (Figs. 2–3), but at the base of the clade that includes Ellobiusini, Lagurini, and Microtini in the mitochondrial analyses (Fig. 1).

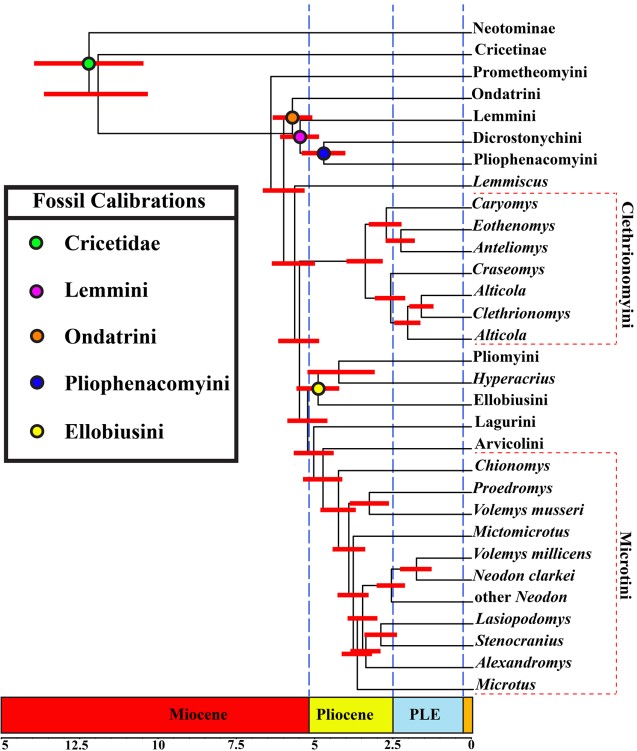

**Figure 4  Time-calibrated phylogeny of Arvicolinae.** Blue vertical dashed lines represent boundaries between geologic epoch. Red horizontal bar at each node represent the 95% HPD for the age of the node. Age in scale bar is in millions of years. PLE =Pleistocene. Orange Box =Holocene. For a species level time-calibrated phylogeny see Appendix S1J.

## Time-calibrated analysis

A tribal-level time-calibrated majority rule consensus tree of the combined dataset is presented in Fig. 4. A species-level tree is presented in Appendix S1J. 105 nodes (72%) had posterior probability values >0.95.

## Divergence-time results

Crown arvicoline rodents were inferred to have diverged ∼6.4 Ma. For a list of all the major clades and their divergence estimates see Table 1.

# DISCUSSION

## Systematic position of genera and discussion of intrageneric relationships

The overall topologies of our ML and BI analyses are largely congruent with previously published molecular phylogenies (*e.g., Conroy & Cook, 1999; Conroy & Cook, 2000; Galewski et al., 2006; Buzan et al., 2008; Fabre et al., 2012; Martínková & Moravec, 2012; Steppan & Schenk, 2017; Upham, Esselstyn & Jetz, 2019; Abramson et al., 2021*) except

**Table 1 Divergence ages of various clades within this study.** Age is in millions of years before present.

| Subfamily | Tribe | Genera | Mean | Median | Minimum age | Maximum age |
|---|---|---|---|---|---|---|
| Neotominae | | | 12.23 | 12.2 | 10.53 | 13.89 |
| Cricetinae | | | 7.86 | 7.85 | 6.5 | 9.27 |
| Arvicolinae | | | 6.41 | 6.4 | 5.7 | 7.17 |
| | Prometheomyini | | 6.41 | 6.4 | 5.7 | 7.17 |
| | Ondatrini | | 3.47 | 3.42 | 3.2 | 3.91 |
| | Lemmini | | 4.05 | 4.02 | 3.95 | 4.23 |
| | | *Synaptomys* | 3.67 | 3.69 | 3.18 | 4.11 |
| | | *Myopus* | 3.11 | 3.11 | 2.62 | 3.61 |
| | | *Lemmus* | 1.65 | 1.64 | 1.22 | 3.09 |
| | Dicrostonychini | | 0.96 | 0.95 | 0.62 | 1.33 |
| | Phenacomyini | | 2.57 | 2.53 | 2.4 | 2.85 |
| | | *Lemmiscus* | 5.65 | 5.64 | 5.06 | 6.3 |
| | Clethrionomyini | | 3.39 | 3.38 | 2.89 | 3.92 |
| | | *Caryomys* | 1.68 | 1.67 | 1.24 | 2.16 |
| | | *Eothenomys* | 1.06 | 1.05 | 0.74 | 1.41 |
| | | *Anteliomys* | 1.83 | 1.82 | 1.47 | 2.18 |
| | | *Craseomys* | 1.45 | 1.45 | 1.08 | 1.83 |
| | | *Alticola* | 2.03 | 2.03 | 1.69 | 2.4 |
| | | *Clethrionomys* | 1.43 | 1.42 | 1.13 | 1.74 |
| | Pliomyini | | 4.2 | 4.23 | 3.15 | 5.17 |
| | | *Hyperacrius* | 4.2 | 4.23 | 3.15 | 5.17 |
| | Ellobiusini | | 3.75 | 3.74 | 3.11 | 4.42 |
| | Lagurini | | 2.92 | 2.91 | 2.33 | 3.51 |
| | Arvicolini | | 1.5 | 1.49 | 1.08 | 1.94 |
| | Microtini | | 4.25 | 4.24 | 3.75 | 4.75 |
| | | *Chionomys* | 2.56 | 2.55 | 2.02 | 3.09 |
| | | *Proedromys* | 3.25 | 3.25 | 2.69 | 3.82 |
| | | *Mictomicrotus* | 3.77 | 3.77 | 3.34 | 4.21 |
| | | *Neodon* | 2.56 | 2.55 | 2.18 | 2.96 |
| | | *Lasiopodomys* | 1.54 | 1.53 | 1.12 | 1.98 |
| | | *Stenocranius* | 1.25 | 1.24 | 0.86 | 1.66 |
| | | *Alexandromys* | 2.15 | 2.14 | 1.77 | 2.51 |
| | | *Microtus* | 3.43 | 3.43 | 3.03 | 3.85 |
| | | NA *Microtus* | 2.92 | 2.91 | 2.54 | 3.3 |

**Notes.**

NA, North American.

for *Robovský, Řičánková & Zrzavý (2008)*, which included morphological characters in their analyses. We did find some different topological arrangements than previous works, especially in relatively earlier divergences, but few of those nodes were well-supported. There are some topological differences based on marker set inclusion (*e.g.*, nuclear, mitochondrial, or combined). Below we outline the implications of our results for the taxonomy and evolutionary understanding of Arvicolinae.

## Basal position of Prometheomyini?

Prometheomyini is sister to all other arvicolines in several studies albeit with weak support (*Galewski et al., 2006*; *Fabre et al., 2012*; *Steppan & Schenk, 2017*; *Upham, Esselstyn & Jetz, 2019*; *Ibiş et al., 2020*). In other studies, it was placed close to the base of Arvicolinae but not as the first diverging arvicoline (*Buzan et al., 2008*; *Robovský, ŘIčánková & Zrzavý, 2008*; *Abramson et al., 2021*). Our nuclear analyses and combined BI analysis placed Prometheomyini as sister to all other arvicolines with strong support (Figs. 2 and 3B). Prometheomyini was deeply nested (though with weak support) in the mitochondrial-only analyses (Fig. 1), and sister to arvicolines besides *Lemmiscus* in the combined ML analysis. Therefore, we stress caution in considering *Prometheomys* as the most basal arvicoline. Discrepancy between the mitochondrial and nuclear loci warrants further investigation (*Pardiñas et al., 2017*; *Kryštufek & Shenbrot, 2022*).

## Systematic relationships within Lemmini

Bog lemmings (*Synaptomys*) and 'true lemmings' (*Myopus*, *Lemmus*) are consistently placed in a clade at or near the base of the arvicoline tree. One previous study placed lemmings as sister to all other arvicolines (*Abramson et al., 2021*), whereas most have found lemmings to be near but not at the base of the tree (*Galewski et al., 2006*; *Buzan et al., 2008*; *Robovský, ŘIčánková & Zrzavý, 2008*; *Fabre et al., 2012*; *Steppan & Schenk, 2017*; *Upham, Esselstyn & Jetz, 2019*). Studies that sampled both species of extant bog lemming found that *Synaptomys* (as defined by *Musser & Carleton (2005)*) is paraphyletic with respect to 'true lemmings' (*Buzan et al., 2008*; *Fabre et al., 2012*; *Steppan & Schenk, 2017*; *Upham, Esselstyn & Jetz, 2019*). Our results unfortunately do not add any clarity on whether bog lemmings should be one or two genera, because 50% of our trees recover a weakly supported clade. However, it should be noted that analyses based either entirely or predominantly on nuclear markers recovered a weakly supported clade, while those that included mitochondrial-only data tended not to. Based on dental morphology, many paleontologists have considered the northern bog lemming, *Synaptomys borealis* (following *Musser & Carleton, 2005*), to be a member of a distinct genus known as *Mictomys* (*Fejfar & Repenning, 1998*; *Repenning & Grady, 1988*). *Musser & Carleton (2005)* and *Pardiñas et al. (2017)* argued that there may be enough evidence to place *Mictomys* and *Synaptomys* as distinct genera, but they tentatively kept them in the same genus. Along with other studies (*e.g.*, *Buzan et al., 2008*; *Fabre et al., 2012*; *Steppan & Schenk, 2017*), our analyses do not support or reject the monophyly of bog lemmings.

## Systematic status of Dicrostonychini and Pliophenacomyini

Voles belonging to *Phenacomys* and *Arborimus* are consistently found to be sister genera (*Robovský, ŘIčánková & Zrzavý, 2008*; *Fabre et al., 2012*; *Steppan & Schenk, 2017*; *Upham, Esselstyn & Jetz, 2019*). Recently, the clade (*Phenacomys*, *Arborimus*) was found to be sister to *Dicrostonyx* (*Galewski et al., 2006*; *Buzan et al., 2008*; *Robovský, ŘIčánková & Zrzavý, 2008*; *Fabre et al., 2012*; *Steppan & Schenk, 2017*; *Abramson et al., 2021*). Historically *Dicrostonyx* was thought to be closely related to the other lemmings, but that was not found in recent studies (*Galewski et al., 2006*; *Buzan et al., 2008*; *Robovský, ŘIčánková*

*& Zrzavý, 2008*; *Fabre et al., 2012*; *Steppan & Schenk, 2017*; *Abramson et al., 2021*).
Our nuclear and combined mitochondrial and nuclear analyses are in agreement with
*Abramson et al. (2021)* that *Dicrostonyx*, *Phenacomys*, and *Arborimus*, are united in a clade
that *Abramson et al. (2021)* called Dicrostonychini. However, we have followed the tribal
terminology of *Kryštufek & Shenbrot (2022)* in which Dicrostonychini (*Dicrostonyx* only)
and Pliophenacomyini (*Phenacomys* + *Arborimus*) are closely related but separate tribes.
Lemmings and ((*Phenacomys*, *Arborimus*), *Dicrostonyx*) were consistently placed in a
clade at or near the base of Arvicolinae in our analyses. Although it is true that *Dicrostonyx*
are not sister to Lemmini, they are part of a more inclusive clade that is often plaed as
sister to Lemmini.

## Systematic status of Clethrionomyini

Using molecular data (and to a lesser extent morphology), *Alticola*, *Anteliomys*, *Caryomys*,
*Clethrionomys*, *Craseomys*, and *Eothenomys* were consistently found to be closely related
and united in the tribe Myodini (*Luo et al., 2004*; *Galewski et al., 2006*; *Lebedev et al., 2007*;
*Buzan et al., 2008*; *Robovský, ŘIčánková & Zrzavý, 2008*; *Fabre et al., 2012*; *Liu et al., 2012*;
*Zeng et al., 2013*; *Steppan & Schenk, 2017*; *Upham, Esselstyn & Jetz, 2019*; *Abramson et al.,
2021*). We use the tribal name Clethrionomyini rather than Myodini, because Myodini
is no longer a valid term (*Kryštufek & Shenbrot, 2022*). Interestingly, dental morphology
alone suggests that *Clethrionomys*, *Craseomys rex* and *Craseomys rufocanus* (rooted
molars) should be more distantly related to *Anteliomys*, *Caryomys*, *Eothenomys*, *Alticola*,
and the rest of *Craseomys* (rootless molars) (*Luo et al., 2004*; *Lebedev et al., 2007*; *Liu et al.,
2012*; *Zeng et al., 2013*; *Kryštufek & Shenbrot, 2022*). However, they are consistently united
together into Clethrionomyini (*Luo et al., 2004*; *Buzan et al., 2008*; *Robovský, ŘIčánková
& Zrzavý, 2008*; *Fabre et al., 2012*; *Liu et al., 2012*; *Zeng et al., 2013*; *Steppan & Schenk,
2017*; *Abramson et al., 2021*). We also found that clade here, and, furthermore, found two
major clades within Clethrionomyini, (1) *Caryomys* + *Eothenomys* + *Anteliomys*, and (2)
*Craseomys* + *Clethrionomys* + *Alticola*.

 *Alticola* was poorly sampled in many previous studies. Depending on which species
of *Alticola* were included, studies variably inferred monophyly (*Fabre et al., 2012*) or
paraphyly (*Lebedev et al., 2007*; *Steppan & Schenk, 2017*; *Upham, Esselstyn & Jetz, 2019*;
*Abramson et al., 2021*) of the genus. Our study includes a robust sampling of currently
recognized species of *Alticola* and our mitochondrial only and combined ML analyses are
similar to *Upham, Esselstyn & Jetz (2019)* in finding a paraphyletic *Alticola* with respect
to *Clethrionomys*. More sampling within *Alticola* is warranted to get a more complete
understanding of this clade.

## Systematic status of *Hyperacrius*

The systematic position of the Subalpine Kashmir Vole (*Hyperacrius fertilis*) has been
relatively understudied. Using morphology alone, *Hyperacrius* was hypothesized to be
closely related to *Alticola* (*Hinton, 1926*), or as a member of the tribe Clethrionomyini
(*Gromov & Polyakov, 1977*). *Kohli et al. (2014)* included *Hyperacrius* for the first time
in a molecular analysis, and its relationship to Clethrionomyini was doubted. We

found weakly-supported close relationships between *Hyperacrius* and Clethrionomyini (mitochondrial only and combined ML) and near Pliomyini (combined BI). Recently, *Hyperacrius* was hypothesized to be the earliest diverging member of what was previously considered Arvicolini (*i.e.,* Microtini minus *Arvicola*) (*Abramson et al., 2020*; *Abramson et al., 2021*). Given that only *cytb* data were available for *Hyperacrius* our results should be considered equivocal and more sampling is needed.

## Systematic status of *Lemmiscus curtatus*

Until recently, the sagebrush vole, *Lemmiscus curtatus*, had not been included in molecular phylogenies (*Steppan & Schenk, 2017*; *Abramson et al., 2021*). Those two analyses produced conflicting results for the systematic position of *Lemmiscus*. Both studies found that *Lemmiscus* and *Microtus* are not sister taxa. *Steppan & Schenk (2017)* placed *Lemmiscus* as sister to *Arvicola*, whereas *Abramson et al. (2021)* placed it as sister to *Chionomys*. Some of our results, though weakly supported, are similar to *Steppan & Schenk (2017)*, probably because we used the same genetic data for *Lemmiscus* for our phylogeny, whereas *Abramson et al. (2021)* used an entire mitochondrial genome of *Lemmiscus*. Our combined ML analysis placed *Lemmiscus* as sister to all other arvicolines whereas our other analyses place it close to Arvicolini. More data from transcriptomes or nuclear genes will help to refine the systematic position of this species, so we do not make any taxonomic recommendations based on the systematic position of *Lemmiscus* in our analyses.

## Systematic status of Ellobiusini, Arvicolini, Lagurini, and Pliomyini

Over the past decade, Ellobiusini, Arvicolini and Lagurini have been the subject of several phylogenetic studies (*Bondareva et al., 2020*; *Mahmoudi et al., 2020*). Ellobiusini (including *Ellobius* and *Bramus*) was placed as an early diverging arvicoline (*Bondareva et al., 2020*; *Robovský, ŘIčánková & Zrzavý, 2008*), or as an early diverging member of the radiation that includes *Lagurus, Eolagurus, Lemmiscus, Neodon, Arvicola, Chionomys, Proedromys, Volemys, Lasiopodomys,* and *Microtus* (*Fabre et al., 2012*; *Steppan & Schenk, 2017*; *Upham, Esselstyn & Jetz, 2019*; *Abramson et al., 2021*). Our results support that Ellobiusini is an early diverging member of the large radiation that includes Arvicolini + Lagurini + Microtini, and not an early diverging arvicoline (see Figs. 1–3). However, that relationship was not strongly supported in any analysis that included mitochondrial data.

Arvicolini has been used to describe a large number of genera including *Arvicola, Chionomys, Neodon, Lasiopodomys,* and *Microtus*. We recognize the tribe Arvicolini as having a single genus, *Arvicola*, based on the *Mammal Diversity Database (2023)*. We inferred a monophyletic Arvicolini with a weak sister taxon relationship to *Lemmiscus curtatus* (in mitochondrial only BI), similar to the results of *Steppan & Schenk (2017)*. That is an interesting biogeographic result given the large distance between the extant members of these genera. We also recovered Arvicolini as the sister clade to Microtini (in both nuclear and combined analyses) and as in *Abramson et al. (2021)*.

Lagurini was placed as the sister to Pliomyini (*Steppan & Schenk, 2017*), as the sister to *Arvicola* (*Abramson et al., 2021*), and as sister to Microtini (*Fabre et al., 2012*). Our

nuclear only and combined results are most similar to *Fabre et al. (2012)* and *Abramson et al. (2021)* in that we found Lagurini to be closely related to Arvicolini and Microtini. Pliomyini was found to be sister to Lagurini (*Steppan & Schenk, 2017*), and as the sister to *Arvicola* (*Fabre et al., 2012*; *Abramson et al., 2021*). Regardless of its sister taxon relationship all three of these studies recovered Pliomyini close to Ellobiusini, Arvicolini, and Lagurini. Our analyses placed Pliomyini as (1) sister to Arvicolini and Clethrionomyini wth mitochondrial data only, or (2) as closely related to Arvicolini, Lagurini, Ellobiusini, and Microtini in the nuclear only and combined analyses. Based on those results, Pliomyini is part of the major radiation that includes Ellobiusini, Lagurini, Arvicolini, and Microtini.

## Systematic status of Microtini

Microtini is the largest tribe of arvicoline rodents. Recently, the *Mammal Diversity Database (2023)* recognized Microtini as distinct from Arvicolini based on *Abramson et al. (2021)*, who found that *Arvicola* was not in a clade with the other members of Microtini. We found the same result (see Figs. 1–3). Therfore we retain the genera *Chionomys*, *Proedromys*, *Volemys*, *Mictomicrotus*, *Neodon*, *Lasiopodomys*, *Stenocranius*, *Alexandromys*, and *Microtus* within a monophyletic Microtini.

Historically, *Chionomys* was placed in *Arvicola*, in *Microtus*, its own genus, or as a subgenus of *Microtus* (*Yannic et al., 2012*). This complicated history can be attributed in part to the fragmented geographic distribution and isolation in high alpine environments of *Chionomys*. *Jaarola et al. (2004)* used analyses of *cytb* to solidify *Chionomys* as a valid genus separate from *Microtus*, and several other studies placed *Chionomys* as a nested member of what is now considered Microtini, but outside of *Microtus* (*Galewski et al., 2006*; *Robovský, ŘIčánková & Zrzavý, 2008*; *Fabre et al., 2012*; *Abramson et al., 2021*). Our results further support that hypothesis and provide evidence to suggest that *Chionomys* is a basal member of Microtini. *Proedromys* has historically been thought to be closely related, or even included in, *Microtus* (*Ellerman & Morrison-Scott, 1951*; *Gromov & Polyakov, 1977*; *Musser & Carleton, 2005*). We concur with other phylogenetic studies (*Fabre et al., 2012*; *Steppan & Schenk, 2017*; *Abramson et al., 2021*) that place *Proedromys* in Microtini and close to but outside of *Microtus*. We consistently found *Volemys musseri* as sister to *Proedromys* with strong support (see Figs. 1–3). Given the geographical overlap of the two genera that result is not suprising and recapitulates previous studies (*Steppan & Schenk, 2017*; *Upham, Esselstyn & Jetz, 2019*). *Mictomicrotus* is a recently named genus that is monotypic (*M. liangshanensis*) and was previously included in *Proedromys* (*Liu et al., 2007*; *Kryštufek & Shenbrot, 2022*; *Steppan & Schenk, 2017*). Our results further support the hypothesis that *Mictomicrotus* is a distinct genus.

Recent systematic and taxonomic work altered our understanding of *Microtus* (*Bannikova et al., 2010*; *Liu et al., 2012*; *Liu et al., 2017*; *Pradhan et al., 2019*). For example, *Neodon* was previously recognized as its own genus or as a subgenus of *Microtus* (see *Musser & Carleton (2005)*; *Pardiñas et al. (2017)*), or was placed in *Pitymys* (*Ellerman & Morrison-Scott, 1951*). It has become increasingly apparent that *Microtus* was historically used as a taxonomic garbage bin. Several species previously allocated to *Microtus* are now

placed in *Neodon* (*N. leucurus*, *N. clarkei*, and *N. fuscus*) (*Pradhan et al., 2019*; *Abramson et al., 2021*). We found a paraphyletic *Neodon* with respect to *Volemys millicens* in most analyses (mitochondrial only ML, nuclear only BI, and combined ML and BI), and a monophyletic *Neodon* that is sister to *Alexandromys* in the others (mitochondrial only BI and nuclear only ML). The relationship between *Volemys millicens* and *Neodon* warrants further exploration.

*Volemys* was considered a distinct genus or a subgenus of *Microtus*, with *Musser & Carleton (2005)* recognizing two species, *V. musseri* and *V. millicens*. That classification is based on morphology alone, and the monophyly of *Volemys* has not been supported by molecular datasets (*Jaarola et al., 2004*; *Steppan & Schenk, 2017*; *Upham, Esselstyn & Jetz, 2019*). We inferred a weakly monophyletic *Volemys* in one analysis (mitochondrial only BI). We suggest further systematic study of *Volemys* and further examination to understand potential morphological homoplasy between *V. millicens* and *V. musseri*.

Voles placed within *Lasiopodomys* and *Stenocranius* have a long history of taxonomic change, with some studies placing them within *Microtus*, while others have placed them all within *Lasiopodomys* (see *Musser & Carleton, 2005*; *Pardiñas et al., 2017*; *Kryštufek & Shenbrot, 2022*). We follow the taxonomy recognized by *Kryštufek & Shenbrot (2022)* and the *Mammal Diversity Database (2023)* in which there are two species of *Lasiopodomys* and two species of *Stenocranius*. All but one of our analyses (mitochondrial only ML) found *Lasiopodomys* and *Stenocranius* to be sister taxa with medium to strong support, in agreement with past studies (*Fabre et al., 2012*; *Steppan & Schenk, 2017*; *Abramson et al., 2021*).

*Alexandromys* was used for holarctic grass voles for more than a century, but its usage has waxed and waned (*Kryštufek & Shenbrot, 2022*). Voles that the *Mammal Diversity Database (2023)* assign to *Alexandromys* have historically been placed in *Microtus*, *Iberomys* (*Gromov & Polyakov, 1977*), *Neodon* (*Musser & Carleton, 2005*), and in various subgeneric groups within *Microtus* (*Kryštufek & Shenbrot, 2022*). Phylogenetic analyses, however indicate the validity of *Alexandromys* as a genus separate from *Microtus*. *Steppan & Schenk (2017)* and *Abramson et al. (2021)* found a monophyletic *Alexandromys* but *Fabre et al. (2012)* found the genus to be paraphyletic with respect to *Volemys musseri*. All of our analyses used a robust sampling within *Alexandromys* and all found a monophyletic genus that is closely related to but outside of crown *Microtus* (see Figs. 1–3). Given the aforementioned issues with *Volemys* it is likely that *Alexandromys* is a monophyletic genus whose recognition helps to make *Microtus* monophyletic.

Voles of the genus *Microtus* are frequently studied but have presented a long-term systematic enigma (*e.g.*, *Conroy & Cook, 2000*; *Fabre et al., 2012*; *Abramson et al., 2021*). *Microtus* is one of the most rapidly evolving lineages of rodents and contains over sixty extant species (*Mammal Diversity Database, 2023*). To attempt to clarify the taxonomy of *Microtus*, researchers have used subgenera such as *Pedomys*, *Alexandromys*, *Terricola*, *Iberomys*, *Agricola*, and *Neodon*, but these subgenera are variably considered genera by different authors. Thus, the genus *Microtus* was in need of being redefined (*Barbosa et al., 2018*; *Abramson et al., 2021*). Recent taxonomic work concerning *Alexandromys*, *Neodon*, *Lasiopodomys*, *Stenocranius*, *Chionomys*, *Proedromys*, and *Mictomicrotus* has

helped to address this issue and our results (a monophyletic *Microtus* in our nuclear only and combined analyses; see Figs. 2–3) further help to support those changes (*Kryštufek & Shenbrot, 2022*). This is a key step to clarifying this complicated genus and getting closer to being able to have an answer for 'What is *Microtus*?'

## Systematic status of North American *Microtus*

The species of *Microtus* that are endemic to North America have in the past been inferred as a clade (*Conroy & Cook, 1999*; *Conroy & Cook, 2000*; *Upham, Esselstyn & Jetz, 2019*; *Abramson et al., 2021*). Our mitochondrial only results conflict with that hypothesis (see species level trees in Appendices S1D and S1G) in that North American *Microtus* is paraphyletic with respect to *M. cabrerae* (a species endemic to the Iberian Penninsula), while our nuclear only and combined analyses (see species level trees in Appendices S1E, S1H and S1F, S1I) found a weakly-supported clade. The relationship between *M. cabrerae* and the North American *Microtus* should be further examined using transcriptome and genomic data. Whether or not North American *Microtus* is monophyletic has important implications for paleobiogeography and our understanding of the fossil record in relation to North American Land Mammal Ages (NALMAs), which is largely based on the immigration of voles and other mammals into North America at various times during the Pliocene and Pleistocene (*Bell et al., 2004*).

## Diversification of arvicolines

Time-calibrated phylogenies rely on several important factors to assure their accuracy and reproducibility. Node calibrations must leverage appropriate fossils to avoid erroneous divergence times. As recommended by *Parham et al. (2012)*, care must be taken to insure that a fossil that is used for a calibration has (1) a museum number, (2) a systematic position established with an apomorphy based diagnosis or phylogenetic analysis, (3) reconciliation of morphological and molecular data with respect to the fossil's position, (4) detailed locality and geological data, and (5) reference to a published radiographic age. Following this protocol insures that a calibration is as accurate as possible, given the understanding of the fossil record at the time of the analysis. Studies have also demonstrated that secondary calibrations can dramatically affect the age of a node (*Schenk, 2016*). Secondary calibrations can certainly be used to help inform node calibrations, but we advise the use of primary fossil calibrations, when possible. In our divergence-time analysis we chose to implement a node-dating approach rather than tip-dating or the fossilized birth-death model (FBD) (*Heath, Huelsenbeck & Stadler, 2014*), because there are currently so few phylogenetically constrained arvicoline fossils. We acknowledge that the use of single fossils for node age minima and maxima (node-dating) is potentially problematic (*Heath, Huelsenbeck & Stadler, 2014*). In the future when there are more phylogenetically constrained fossils, we recommend a reexamination of this issue using FBD methods.

We argue that one of the strengths of our paper is our justification of our fossil calibrations following *Parham et al. (2012)*. Our results are largely congruent with the published fossil record and our divergence time estimates are younger (closer to the

known fossil record) than those produced by *Steppan & Schenk (2017)*, *Upham, Esselstyn & Jetz (2019)*, and *Abramson et al. (2021)*. This is not surprising given the density of fossil calibrations that we used across the arvicoline tree. We estimated a median age of crown Arvicolinae at ~6.4 Ma, which is slightly younger than the age (7.4 Ma) inferred by *Abramson et al. (2021)*. These ages are very close to the age of the earliest known possible arvicoline rodent fossils (~9–8 Ma, which are probably Pliocene in age (*e.g.*, *Pannonicola*, *Microtoscoptes*, or *Goniodontomys*; *Maul et al., 2017*). *Abramson et al. (2021)* included a calibration at the crown Arvicolinae node (7 Ma FAD, 10 Ma max age) whereas we did not, potentially accounting for some discrepancy in our results. We did not use a calibration at the root of Arvicolinae because of the uncertain phylogenetic placement of putative early arvicoline fossils (*Repenning, 1987*; *Fejfar et al., 2011*). Thus we hypothesize an origin of arvicolines c. 6.4 Ma, with dentally distinct arvicoline rodents evolving slightly later. For ages of major tribes and clades of arvicolines found in our study, see Table 1.

## CONCLUSIONS

A consolidated understanding of the phylogeny of arvicoline rodents has been warranted given their remarkable evolutionary history and abundance across high latitudes. We provide phylogenetic support for systematic hypotheses across Arvicolinae and some direction for future systematic and taxonomic work. We show that the earliest diverging arvicolines likely includes the Tribes Prometheomyini, Ondatrini, Lemmini, Dicrostonychini, and Pliophenacomyini. *Prometheomys* is probably the sister taxon of all other arvicolines, but work still needs to be done to solidify that hypothesis. The monophyly of bog lemmings (*Synaptomys*) is doubted, however their close relationship to the "true lemmings" (*Myopus* and *Lemmus*) is clear. *Hyperacrius* and *Lemmiscus* need to be sampled using phylogenomic data to better clarify their systematic positon. Clethrionomyini is a valid Tribe consisting of *Caryomys*, *Eothenomys*, *Anteliomys*, *Craseomys*, *Alticola*, and *Clethrionomys*. The paraphyly of *Clethrionomys* relative to *Alticola* needs to be explored. Pliomyini, Ellobiusini, Lagurini, Arvicolini, and Microtini represent a clade that forms a large portion of the arvicoline tree. We found some evidence for a monophyletic endemic North American *Microtus*, however this was not found in all analyses, and could benefit from larger scale sampling across the genome. Finally, we estimated divergence times among the major clades, which were concordant with the published fossil record. This will provide valuable insight into the evolutionary and paleobiogeographical history of this clade. Overall our understanding of the evolutionary history of arvicolines is improving and this study is a step in the right direction for both taxonomic and systematic clarity for Arvicolinae.

## ACKNOWLEDGEMENTS

We are grateful to the PhD Committee of C. Withnell whose advice and comments helped shape this manuscript. This committee included C. Bell, C. Jass, D. Cannatella, T. Rowe, and R. Martindale. We would also like to thank the members of the Bell Research

Lab at The University of Texas at Austin whose support was much appreciated. Having not generated any new molecular data for these analyses, we are immensely grateful to the many researchers whose work made this synthesis possible (see Appendix S1A for references to all authors who contributed data used in this study). We are grateful to Kenneth De Baets (editor), Connor J. Burgin (reviewer) and two anonymous reviewers for providing suggestions that greatly improved the paper.

### Funding
The authors received no funding for this work.

### Competing Interests
The authors declare there are no competing interests.

### Author Contributions
- Charles B. Withnell conceived and designed the experiments, performed the experiments, analyzed the data, prepared figures and/or tables, authored or reviewed drafts of the article, and approved the final draft.
- Simon G. Scarpetta conceived and designed the experiments, performed the experiments, analyzed the data, authored or reviewed drafts of the article, and approved the final draft.

### Data Availability
The data is available on Dryad: Withnell, Charles; Scarpetta, Simon (2023). A new perspective on the taxonomy and systematics of Arvicolinae (Gray, 1821) and a new time-calibrated phylogeny for the clade [Dataset]. Dryad. https://doi.org/10.5061/dryad.qrfj6q5cg.

### Supplemental Information
Supplemental information for this article can be found online at http://dx.doi.org/10.7717/peerj.16693#supplemental-information.

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
