# Peer review of "A new perspective on the taxonomy and systematics of Arvicolinae (Gray, 1821) and a new time-calibrated phylogeny for the clade"

_PeerJ, doi:10.7717/peerj.16693_

## Round 0.1 · original submission · Major Revisions

Apologies for the delay in making my decision but given the disparate reviews, I needed time to go through the reviews and the manuscript back-to-back in detail. You provide a new time-calibrated phylogenetic analysis using all existing relevant sequences of 89% of Arvicolinae as well as provide a timely review of the existing literature on taxonomy and phylogeny of this group. I agree with most reviewers that such an endeavor is worthy contribution to the field and would like to see it published. However, some crucial points need to be addressed before publication:

1) More detailed designation and description of voucher specimen: you need to more clearly document the source of the sequences used (e.g., the museums and specimens for which they came). Also please provide designation and description of the available voucher specimens (compare reviewer 1). Providing this information is critical for the sake of scientific reproducibility and confirming the species identities. Uncertainties surrounding their identities might also contribute to discrepancies between your analysis and other studies. I agree with reviewer 1 this could be achieved by re-organizing the table that lists GenBank numbers and adding information on the authors who published those data and, if available, the specimens from which those DNA sequences were derived, museum names and museum catalog numbers. The table would also benefit from an alphabetic ordering of sequences according to the genus/species name.

2) Used sequences: I agree with reviewer 1 that as much as possible, vouchered sequences should be used and other sequences should be treated with caution - also the latter should be more clearly designated. I also agree with reviewers 1 and 3 that the study could be of even stronger and of greater relevance when new sequences are added (e.g., to fill gaps in the used markers) but I also understand this might be potentially beyond the scope of your study. At minimum, the analysis will need to be re-run after a higher degree of scrutiny as available sequences of Neodon are missing or wrongly assigned, those of Volemys milicens likely belong to N. clarkei and includes additional problematic sequences (see reviewer 3). Please also provide a range of the completeness of the data from the minimum (the species with smallest amount of data) to maximum (species with complete data; compare reviewer 1).

3) Scope of the study: The scope should be stated more clearly in the study. As succinctly stated by reviewer 3: What was the objective of the paper or why a new phylogeny for Arvicolines is needed? See also reviewer 1 about misunderstanding the scope of your study.

4) Analysis of mito-nuclear discordance: I agree with reviewer 2 that the use of 5 genes is needed. However, the focus on the concatenation of the nuclear and mitochondrial genes on its own might be the wrong way to go and I agree with the recommendations by reviewer 2 to continue the include the concatenated mtDNA + nuclear genes, but to put more focus on the nuclear and mitochondrial trees separately. This would make your study of broader relevance and more clearly highlight the potential limitations of it.

5) Divergence time analysis: It is not clear why MrBayes was used while novel analyses in Beast (or RevBayes) likely can deal better with differences in rates between nuclear and mitochondrial genes (compare reviewer 1) as well as the uncertainty surrounding fossil calibrations.

6) Rerunning analyses with different combination of genes: I agree with reviewer 3 that there is no need to retain those analyses in main text as model-based methods for constructing phylogenetic trees are relatively robust to missing data, so excluding genes based on missing data is not necessary.

7) Use of secondary rates: you explain the reason behind using secondary rates for calibration of one node, but the pitfalls of using such an approach should also be stated more clearly (see for example Schenk 2016). Also, you clearly need to state the clock rates in terms of substitutions per site per some unit of time (compare reviewer 1).

8) Biogeography: Although biogeographic history is alluded to throughout the text, there are no clear biogeographic hypotheses stated nor are biogeographic implications discussed in greater detail. I feel adding a larger biogeographic component to your study would greatly increase its relevance (compare reviewers 2 and 3). At minimum the biogeographic history of the group should be explained in detail which might explain the radiation events (compare reviewer 3). For example (compare reviewer 2): How does your tree fit into the context of more recent advances in arvicoline systematics and Holarctic biogeography from the Miocene to the present? You would be in a unique position to do this as studies only rarely analyse taxa across both the Palearctic and Nearctic/Neotropics.

9) Recovered radiations: you identify 4 radiations as opposed to 3 in some previous analyses (e.g., Abramson et al. 2021). However, the first “radiation” does not display a rapid increase in speciation events as seen in events 2-4 (compare reviewer 3). You need to justify why you also consider this event as a radiation event.

10) Literature review of classifications of Arvicolinae: you already provide a substantial review of the literature, and all reviewers consider it as one of the main strengths of your study in addition to the divergence time estimates. Some additional references could be considered for the sake of completeness (compare reviewers 1-3). Particularly the inclusion of more updated classifications of Arvicolinae (see recommendations by reviewer 2) are crucial to add the modern context to your analysis. I agree with reviewer 2 that the use of the 3rd Edition of the Mammal Species of the World (MSW3) for the initial taxonomic arrangement used for comparison might be a bit outdated. Your analyses and particularly the discussion would be more updated and enriched by focusing on they support the already proposed change as catalogued in the American Society of Mammalogists Mammal Diversity Database (MDD; mammaldiversity.orgy) as well as other compendia such as the 7th volume of the Handbook of the Mammals of the World, Rodents II, and the Arvicolinae of the Palearctic by Boris Kryštufek and Gregory Shenbrot which are also included in the MDD (compare reviewer 2)

11) Figures: I agree with reviewer 3 that figures 1 and 3 can be combined with BS and PP placed in the nodes togethe and that figures 2, 4, and 6 can be moved to the supplemental section. It would be crucial to present the trees as a majority rules consensus tree with any node with support values BS<50 or PP<0.50 collapsed into a polytomy (compare reviewer 3). See also recommendation by reviewers 1 and 2 concerning the presentation of the trees. Please make sure the nodal support values are legible and/or use other ways to designate if support values fall within a certain range (compare reviewer 1). Please add the Holocene (or alternatively Quaternary = Pleistocene + Holocene) to the timeline and make sure the legend is correct in figures 5 and 6 (compare reviewer 1).

Please make sure these points as well as all other points raised by the reviewers and me including those in annotated pdfs are addressed.

I look forward to receiving your revised manuscript.

Suggested reference:
Schenk, J. J. (2016). Consequences of secondary calibrations on divergence time estimates. PloS one, 11(1), e0148228.

Reviewer 1 ·

Basic reporting

The manuscript could do a better job explicitly acknowledging the contributions of the different entities that made it possible for them to use the data that they acquired. For example, the supplemental table that lists the GenBank numbers should also include information on the authors who published those data and, if available, the specimens from which those DNA sequences were derived. This should include museum names and museum catalog numbers, which is critical to ensuring the reproducibility of the data and, potentially, confirming species identities. Further, documenting the specimens from which the data came ensures that museums can track the impact of their collections in supporting current research. If the DNA sequences were not vouchered with archived specimens, they should be denoted as such and these should be viewed with some caution as there is no way to independently verify their identity. In cases where multiple sequences are available on GenBank, but not all have archived vouchers, the vouchered sequences should be used. The table should also be organized in a predictable way – for example, alphabetically by genus/species. Currently there is no clear organizational scheme, so finding data for a specific taxon requires a bit of hunting around.

The main strength of the paper is in its clearly written and largely comprehensive treatment of the relevant literature, which is by no means straightforward to review. Indeed, I would argue that the major contribution of the study as written is its review of the literature rather than its phylogenetic results. I only noted a few places that might merit inclusion of an additional source. For example, in discussions of lemmings (lines 83 to 96), perhaps consider addressing work by Vadim Fedorov (doi.org/10.1046/j.1420-9101.1999.00017.x), and in discussions of Alticola and Clethrionomys (e.g., line 594) consider how Brooks Kohli’s work (which is cited elsewhere in the paper) compares to these results. It would also be appropriate to consider Andrey Lissovsky’s work on Alexandromys voles (DOI: 10.1111/zsc.12261).

Experimental design

This manuscript attempts to contribute to clarifying the taxonomy and systematics of the arvicoline rodents. This is an admirable goal, as it is certainly the case (as is pointed out by the authors) that many questions remain despite decades of investigation by numerous research groups. However, I was very disappointed to see that in this study, despite references to “our data” (e.g., line 573), the authors did not present any new data of their own to help resolve the ambiguities that remain in the arvicoline phylogeny. Instead, the authors simply extracted existing sequences from GenBank and re-analyzed them. Certainly it is possible that such an approach could yield new insights if the data in question had been collected and submitted to GenBank for purposes other than phylogenetic analyses, but this does not appear to be the case here. That is evidenced by the fact that most of the phylogenetic observations reported here (both robust results and weak results) were previously described by the authors who published the sequence data in the first place. I found only one instance of an apparently robust and unexpected result that differed from prior published work (see line 605 regarding monophyly of Alticola). Further, given that the authors did not contribute any new data to the problem at hand, their repeated recommendations that new phylogenomic data should be collected ring a bit hollow.

I encourage the authors to consider how they could truly help advance the field by collecting new data. While large genomic datasets would be great, I don’t think that the bar for making a meaningful contribution must be so high. For example, many of the species of Microtus (one of the most problematic groups) are not represented by all 5 genetic markers that were used in this study. Several are only represented by a single gene. It would potentially be enlightening to fill those holes in sampling by sequencing the missing markers from taxa with incomplete data. This might require developing collaborations with research groups that can provide the necessary tissues or DNA extracts, of course.

Validity of the findings

The methods employed and associated results and conclusions of the study are for the most part sound. The one major caveat to this is the following:

L400-401 – While the majority of the analyses seemed to be carried out appropriately, the molecular clock raised some questions. First, clock rates are generally written in terms of substitutions per site per some unit of time, but here there is no indication of the time over which 0.08 substitutions/site will accrue. Is this per million years (which is common and looks about right)? Also, am I correct that the 0.08 subst/site/time rate was calculated for mitochondrial DNA specifically? It is exceedingly unlikely that the same rate would apply to nuclear data. If the structure of the analysis was such that this clock rate was used to inform divergence times based on nuclear sequence data (which I believe was the case), this needs to be corrected. If this analysis was completed using BEAST (which might be a good idea as it was built to do exactly the kind of analysis that is being done here), the rate can be applied only to the mitochondrial data, but it isn’t clear that MrBayes allows such granular control.

Additional comments

Line 241 – “Sequences… were compared to other sequences…” It is not clear what this comparison was meant to accomplish, or how it was interpreted. One can compare a sequence to others in GenBank using BLAST, and though you will usually get a hit, there is no guarantee that the database sequence and the query sequence belong to the same species. What criteria were applied to draw conclusions, if this is the intent behind this comparison?

L244 – While species with complete data might have a total of 5220 bases, that is not the case for most of the species. Perhaps give a range from minimum (the species with smallest amount of data) to maximum (species with complete data).

L455 and elsewhere – The use of the word “basal” to mean “derived from the basal node” is (though widespread) inaccurate and to be avoided. No species alive today is basal; rather, the common ancestor of a clade is basal.

L455-498 – It is not always clear which tree is being referenced when specific support values are reported. In at least one place where the specific tree was referenced (L492), the support value that is cited (1.00 PP in Fig. 4) does not appear to be correct for that tree (to the extent that I could read the support values in the relevant section of the tree – see below).

L617 – What is meant by “our analyses recovered Hyperacrius”? It recovered a specific relationship between Hyperacrius and something else?

L673 – “Our results… further substantiate the claim…” This is only true if you are providing new data. Reanalyzing old data that were used to propose a hypothesis does not provide a test of that hypothesis.

L762-776 – By what criteria are these “radiations” determined? I tend to think of a radiation as being indicated by a clade that shows a burst of speciation in a narrow window of time, but not all of these numbered “radiations” show such a pattern. Do the node age estimates play a role here?

Figures 5 and 6 – The timeline on the bottom is missing the Holocene. If you want to not include it, you should change “Pleistocene” to “Quaternary” (which includes both the Pleistocene and the Holocene). Also, in the legend, the node bars are not confidence intervals (given that they come from a Bayesian method). They are HPD (highest posterior density) intervals. They serve a similar function to confidence intervals.

All tree figures – Nearly all nodal support values were illegible in the review version of the manuscript. Presumably a published figure will have a higher resolution, but regardless, the font was microscopic. I am aware of the challenges of putting support values on a tree with a lot of branches, but making the font so small that one must pull out a magnifying glass doesn’t seem like a good strategy. Many papers will use symbols on branches to denote support values that fall within a certain range. Consider how to make these trees more readable.

·

Basic reporting

Overall, the phylogenies presented in this paper provide further support for many of the taxonomic arrangements recently proposed for Arvicolinae, which despite the small number of genes used is a great step in systematics of arvicolines given that few papers have provided a focused and (mostly) complete analyses of the genera across both the Palearctic and Nearctic/Neotropics. However, they're use of the Mammal Species of the World 3rd Edition (MSW3) as the initial taxonomic arrangement used for comparison was probably not the best route to take, especially since the generic and tribal taxonomy within Arvicolinae has experienced a number of substantial changes. Many of the changes since 2005 (almost 20 years of research now) already address many of the inconsistencies they've noted within Arvicoline taxonomy in their Discussion and they did not include mention to the the publications that made these changes in their taxonomic history section. I'd like to see them address these changes and rather than suggestion they be made, discuss that they found further support for these already proposed changes (e.g., the inclusion of Blandfordimys under Microtus, the placement of gregalis and raddei under Stenocranius, the recognition of Alexandromys as separate from Microtus, etc.; many of these changes have become the standard in arvicoline systematic research). Most of these changes have been catalogued in the American Society of Mammalogists Mammal Diversity Database (MDD; mammaldiversity.orgy), which includes brief descriptors on what's changed since MSW3 and provides the appropriate references for each taxonomic change. Two other compendia sources that would be good to reference would be the 7th volume of the Handbook of the Mammals of the World, Rodents II, and the Arvicolinae of the Palearctic by Boris Kryštufek and Gregory Shenbrot (https://library.oapen.org/handle/20.500.12657/57647), although both sources have been implemented in the taxonomic listing of the MDD. Before this paper is published, I'd like to see a more comprehensive analyses of the current taxonomic standing within Arvicolinae and how their phylogenies fit into the context of meodern systematics rather than on the outdated taxonomy of MSW3, especially given the considerable amount of research that has gone into the current systematic arrangement of subfamily. To pass in my opnion, I'd expand further on the modern phylogenetics associated with each of the genera and compare your work to the arrangement listed below (references shown on the MDD and other associated works). Otherwise, I do think this paper has potential to be relevant to modern arvicoline systematics, but it needs to be placed in the context of modern works in association to the older taxonomy they review (which is done well in my opinion).

I'd also like to point out that some of the systematic history of Arvicolinae can be used to answer broader questions in biogeography, which is poorly presented in the texts discussion. I'd suggest adding more information on the biogeographic history associated with the split of the various clades and the rapid radiation of microtine voles in particular. Maybe some discussion on whether arvicolines could be considered an adaptive radiation based on the diverse ecologies and morphologies within the group (although there is considerable morphological conservatism in microtines in particular)?

The MDD taxonomy includes the following taxonomic arrangement with 169 species (although see Kryštufek and Shenbrot's book for a different tribal/subtribal arrangement, which I do not personally endorse do to the heavy use of subtribes, which are rarely used in modern systematics):

Subfamily Arvicolinae
Tribe Arvicolini (Arvicola 4 sp)
Tribe Clethrionomyini (Alticola 14 sp, Anteliomys 7 sp, Caryomys 2 sp, Clethrionomys 5 sp, Craseomys 5 sp, Eothenomys 3 sp)
Tribe Dicrostonychini (Dicrostonyx 7 sp)
Tribe Ellobiusini (Bramus 2 sp, Ellobius 2 sp)
Tribe Lagurini (Eolagurus 2 sp, Lagurus 1 sp)
Tribe Lemmini (Lemmus 3 sp [RECENT 2022 REVISION CHANGED BACK TO 6 SP, see Abramson et al., 2022], Myopus 1 sp, Synaptomys 2 sp)
Tribe Microtini (Alexandromys 13 sp, Chionomys 5 sp, Hyperacrius 2 sp, Lasiopodomys 2 sp, Lemmiscus 1 sp, Microtus 60 sp, Mictomicrotus 1 sp, Neodon 10 sp, Proedromys 1 sp, Stenocranius 2 sp, Volemys 2 sp)
Tribe Ondatrini (Neofiber 1 sp, Ondatra 1 sp)
Tribe Phenacomyini (Arborimus 3 sp, Phenacomys 2 sp)
Pliomyini (Dinaromys 2 sp)
Prometheomyini (Prometheomys 1 sp)

Reference to recent Lemmus work:
Abramson, Nataliya, Tatyana Petrova, and Nikolay Dokuchaev. "Analysis of “historical” DNA of museum samples resolve taxonomic, nomenclature and biogeography issues: case study of true lemmings." Biological Communications 67.4 (2022): 340-348.

Experimental design

Given the scope of the project, I think the use of a 5 genes is warranted, especially since molecular sampling across the subfamily is sparse in many groups. I also have no qualms with how the trees were generated and the use of fossil calibration to estimate divergence times (the fossils used in calibration were well thought out). However, the concatenation of the nuclear and mitochondrial datasets seems misplaced, especially given the evidence of mito-nuclear discordance found in many mammalian clades and that mitochondrial genes tend to 'swamp out' nuclear genes when concatenated. I would suggest continuing to include the concatenated mtDNA + nuclear genes, but would put more focus on the nuclear and mitochondrial trees separately. The concatenation of mitochondrial and nuclear genes is generally being moved away from in modern phylogenetics because of the discordance in gene trees. I would heavily suggest creating a figure comparing the mtDNA and nuclear trees for mit-nuclear discordance, which would provide further context to the potential for ancient and modern introgression between clades and species, respectively. Having this dataset would put add to how citable your paper is, as it would lay out any issues in the gene trees your using. Also, I would add more context on the limitations of your study. Since you are using only 3 nuclear genes, they don't provide a robust framework to estimate species tree topology and that mtDNA is a good proxy for time calibration given the relatively consistent mutation rate (opposed to using multiple nuclear markers), but does only provides a phylogeny based on maternal inheritance and can lead to incomplete views of the evolutionary history of clade given mitochondrial capture and unidirectional introgression.

Validity of the findings

This certainly is meaningful and valuable research within the arvicoline community and does have broader ramifications for the biogeographic history of Holarctic mammals through the Miocene and Pleistocene, but I would say that they need to be more clear with how there research fits into broader questions in biogeography and modern arvicoline taxonomy (as discussed in the 1st section). I think the conclusions need to be compared to modern arvicoline taxonomy to make their findings more relevant to the greater mammalogical community. They call for further research to establish a firm systematic arrangement within Arvicolinae, but make suggestions that have already been discussion and accepted, thus restating these concepts as support for earlier taxonomic change would be more valuable. Otherwise, I think the authors are on the right track and just need to update the text to reflect newer arrangements.

Additional comments

I think this is a great contribution and am excited to see it published, but it needs a good bit of work in the text and may be more citable with an analysis of mito-nuclear discordance and focusing on how their tree fits into the context of more recent advances in arvicoline systematics and Holarctic biogeography from the Miocene to the present. Most of the edits to make would be in the taxonomic background and discussion, as well as the figures to reflect a more modern taxonomic arrangement.

A minor issue I didn't think fit into the above sections well is how the phylogenies are presented. It may be good to show at least some of the trees summarized by genera rather than nodes down to the species for all of them. Just aids in readability, but I don't think it's completely necessary and if it's to much of a hassle, the trees do adequately present the tree topology in a readable (if not easily readable) format.

Reviewer 3 ·

Basic reporting

No comment

Experimental design

This paper consolidates the available sequences for Arvicolines, a very difficult group to resolve phylogenetically. No sequences were generated to construct the phylogeny and hence there is relatively little new inferences that can be made from the phylogenetic analyses. The phylogenetic relationships, while not in complete agreement of any published phylogeny, do not lead to any new understanding between the major groups. The strength of the paper lies in the excellent summary provided about the taxonomic history of the group and the dated time calibrated phylogeny, but little inference about the biogeographic history of the group is made. In the current state, the paper would require major revisions before publication.

The paper would be strengthened by:
1. Inclusion of a clear reasoning as to why the paper is needed. What was the objective of the paper or why a new phylogeny for Arvicolines is needed?
2. Discussion of biogeographic history of the group that would explain the radiation events.

Validity of the findings

The paper needs to have analyses redone with additional data and with some problematic data removed. The revisions needed are broken into major and minor categories below:

Major problems:
The analyses will have to be rerun with new sequences for:
1. Neodon is under-represented in the analyses. Sequences are available for other species.
2. Neodon leucurus - AM392371, AM392394, AM919400 likely belonged to N. fuscus
3. Volemys milicens sequences included are problematic and belong to N. clarkei.
4. For more problematic sequences detected, please refer to Pradhan et al. (2019)
Figures 1 and 3 can be combined with BS and PP placed in the nodes together. Figures 2, 4, and 6 should be moved to the supplemental section. The trees should be presented as a majority rules consensus tree with any node with support values BS<50 or PP<0.50 collapsed into a polytomy.
Radiation events 2,3, and 4 appear to be radiation events, however event 1 does not display a rapid increase in speciation events like events 2, 3, and 4. Need to justify why event 1 is considered a radiation event, when it was not in Abramson et al. (2021).
Analyses were rerun with different combination of genes. Need to compare the trees. In the current state, there is not enough justification to retain these analyses. Maybe move them to the supplementary section. Model based methods for constructing phylogenetic trees are relatively robust to missing data, so excluding genes based on missing data is not necessary. You may try to get a species trees with *BEAST from gene trees, if that is of interest.

Minor problems:
Line 216 – number of species needs revision. Refer to the newly publish book by Krystufek and Shenbrot (Voles and Lemmings Arvicolinae of the Palearctic region)
Line 495 – clade is not weakly supported if BS<50 or PP<0.50. This means no support.
Line 502 – the time-calibrated majority rule consensus tree is more likely a maximum clade credibility tree
Line 586 – Jin et al. (2013) not in references section
Line 725 – Fabre et al. (2004) not in references section
Lines 859-867 – need to label which of these are a and b as cited in the text Line 239
Line 1172 – Wilson paper not cited in the text

Additional comments

No comment

---

## Round 0.2 · Minor Revisions

The manuscript provides a thorough, timely review and analysis of the arvicoline evolutionary history. I apologize for the delay in my decision, but I was hoping to receive at least one additional review. The revisions make it easier to follow, to reproduce and make it a substantive contribution to the field. Particularly the effort taken to properly acknowledge the data sources as well re-organization of the text around the arvicoline tribes will be highly appreciated by the community and readers. There are still some minor, but crucial points I would like to see addressed before publication. These include:

1) Concatenation: Splitting mitochondrial and nuclear data into separate and joint analyses was a good decision, but the rationale behind concatenating nuclear loci might not be fully justified either. This approach needs to be at least better justified and potential pitfalls (e.g., impact on unresolved relationships in tree) discussed (compare reviewer 1)

2) Substitution model selection: As the validity of your analyses rely on it, I agree with reviewer 1 that it is crucial to justify and explain better the (quantitative) selection of nucleotide substitution models and clarify the rationale behind their application to specific partitions in phylogenetic analyses. The most critical points are discussed at greater length by reviewer 1.

3) Divergence time estimation: I agree with the reviewer that strength of your analysis lies in the thorough discussion and justification of fossil calibrations. The use of simpler divergence time estimation approach is ok, but its potential caveats and perspectives for future analysis should be mentioned and discussed in greater detail (see also reviewer 1)

4) Mutation rate: I agree with reviewer 1 that rate cited in line 447 is not too far outside the norm for rates estimated from small mammals. The manuscript would benefit from clarifying this point and/or potentially stating it more explicitly supported by references. Alternatively, this point could potentially also be deleted (compare reviewer 1), although I would prefer the first option.

5) Figures: Please clarify and resolve confusion surrounding tips in tree figures that are labelled only as genera versus those branches with species names (compare reviewer 1). The circles relating to the fossil calibration in Figure 4 are too large as they sometimes blur underlying branches. Please consider making them slightly smaller (e.g., more in line with other circles in other figures).

6) Appendix A: the appendix provides a key reference for the original data sources, but please make sure a legend or proper explanations are presented which allow the reader to understand its structure and designations (compare reviewer 1)

Please make sure to address these and other points raised by the reviewer and myself (see also annotated pdf).

I look forward to receiving your revised manuscript.

Reviewer 1 ·

Basic reporting

This manuscript is significantly revised relative to the original submission. These revisions have improved the manuscript considerably. Organizing the Introduction and Discussion around the arvicoline tribes, while making for somewhat stilted reading, was a good decision as it helps the reader quickly find information on particular groups that are of interest to them. As was the case in the original draft, the major contribution here is in the review of the relevant literature, which will be a useful resource for investigators in the field.

Appendix A does a fine job of addressing the need to give credit to the original data providers, though there is no legend that describes what is in the table. For example, some cells are left empty, and others say “NONE”. The distinction between these is unclear. I assume that “NONE” is intended to indicate that no voucher was available for a specific sequence, while being left blank means that there was no sequence for that marker/taxon, and therefore no voucher. However, at least one cell rejects that hypothesis (N5). It would be worth going through this one more time to ensure that it is fully correct.

Experimental design

Splitting the mitochondrial and nuclear data into separate and combined analyses was a good decision, although concatenating the nuclear loci still suffers from the same problems that concatenating mitochondrial and nuclear data suffers from (i.e., independent assortment potentially leading to conflicting phylogenetic signal among loci). Perhaps it is reasonable to assume that most internodal distances deep in the phylogeny are long enough for lineage sorting to have completed at all loci, but it seems unlikely that this is necessarily the case for all internal branches. It seems likely that rapid lineage splitting and failure to complete lineage sorting could explain some of the more stubbornly unresolved relationships in the tree, and it would be reasonable to provide some discussion about this.

Decisions regarding selection of nucleotide substitution models need to be better explained. As described, it appears that models were quantitatively selected for only the nuclear data, and even that is unclear as the nuclear loci were evidently partitioned by codon position, yet only 2 models were selected (and it is unclear to which partition they were applied). Why were the mitochondrial sequences not partitioned? Why were substitution models not assessed quantitatively for the mitochondrial data? If they were, then this should be explained. It is likely that partitioning by codon position would be even more important for the mitochondrial data than for the nuclear data given the elevated rate of molecular evolution at mitochondrial loci. This is especially the case for 3rd positions, which could very well be approaching saturation over the relatively long timeline evident in the phylogeny. Assuming that model selection is fully addressed, the application of these models to specific partitions in specific phylogenetic analyses needs to be clarified.

The clock-calibrated analysis makes a contribution, though its conclusions require the caveats that all such analyses require. That said, the clear explanation and justification of the fossil calibrations is a strength of this analysis.

Validity of the findings

The conclusions appear to be appropriate for the results and correctly described.

Additional comments

Line 447 – The mutation rate cited here is not really too far out of the norm for rates estimated for many small mammals, so stating that it is “substantially higher than many if not most other
448 mammals” doesn’t strike me as true. It’s also not clear that it is a necessary statement. Also, note that “i.e.” means “that is”. Use of “i.e.” here implies that Pan, Bos, and Ursus represents “most other mammals”. It would be more appropriate to use “e.g.”, which means “for example”. This issue showed up in a couple places.

The inclusion of tips in tree figures that are named using both genus-only and full species names is confusing. For example, several trees have a branch labeled Clethrionomys and other branches with species names such as Clethrionomys rutilus and Clethrionomys gapperi. The branch with just the genus name implies that all members of the genus are on that branch, but that is obviously not the case. Perhaps indicate “other Clethrionomys”, if that is the intent?

---

## Round 0.3 · accepted · Accept

Thank you for addressing our final suggestions. The addition of additional information concerning concatenation, choice of substitution models, pitfalls and perspectives of divergence time estimation and mutation rate, which makes the manuscripts even easier to follow, reproduce and of broader relevance. The smaller circle of fossil calibrations make it easier to see the underlying branches. I found no new formatting or other issues in the new version. I look forward to seeing this work published.